# On the calculus of smoothing kernels for seismic parameter spatial mapping: methodology and examples

David Montiel-López[1], Sergio Molina[1, 2], Juan José Galiana-Merino[3, 4], and Igor Gómez[1, 2]

[1]Multidisciplinary Institute for Environmental Studies (IMEM), University of Alicante, Crta. San Vicente del Raspeig, s/n, 03080 Alicante, Spain
[2]Department of Applied Physics, University of Alicante, Crta. San Vicente del Raspeig, s/n, 03080 Alicante, Spain
[3]University Institute of Physics Applied to Sciences and Technologies, University of Alicante, Crta. San Vicente del Raspeig, s/n, 03080 Alicante, Spain
[4]Department of Physics, Systems Engineering and Signal Theory, University of Alicante, Crta. San Vicente del Raspeig, s/n, 03080 Alicante, Spain

**Correspondence:** David Montiel-López (david.montlop@ua.es)

**Abstract.** Spatial mapping is one of the most useful methods to display information about the seismic parameters of a certain area. As in b-value times series, there is a certain arbitrariness regarding the function selected as smoothing kernel (which plays the same role as the window size in time series). We propose a new method for the calculus of the smoothing kernel as well as its parameters. Instead of using the spatial cell–event distance we study the distance between events (event-event distance) in order to calculate the smoothing function, as this distance distribution gives information about the event distribution and the seismic sources. We examine three different scenarios: two shallow seismicity settings and one deep seismicity catalogue. The first one, Italy, allows to calibrate and showcase the method. The other two catalogues: Lorca (Spain) and the Vrancea county (Romania) are examples of different function fits and data treatment. For these two scenarios, the prior-earthquake and after-earthquake b-value maps depict tectonic stress changes related to the seismic settings (stress relief in Lorca and stress build-up zone shifting in Vrancea). This technique could enable operational earthquake forecasting (OEF) and tectonic source profiling given enough data in the time-span considered.

## 1 Introduction

The ultimate goal of operational earthquake forecasting (OEF) is to disseminate authoritative information using short-term time-dependent seismic hazard to help communities to prepare and manage damaging seismic emergencies. On the other hand, earthquake prediction pretends to determine the future occurrence of a given earthquake from the observable behaviour of earthquake-related parameters. Both approximations have to be given with corresponding uncertainties if the information is going to be used for further preventive action. Currently, the scientific community agrees with the hypothesis that the use of precursors has not yet provided a short-term seismic prediction framework (Uyeda and Nagao, 2018). However, in the ongoing research there is optimism on using OEF (Martinelli, 2020). Additionally, the integration of different seismic precursors in the OEF analysis might improve the reliability of the obtained results and help to improve the development of earthquake prediction systems in the future.

A seismic precursor is a phenomenon that takes place prior to the occurrence of an earthquake. There are several precursors, classified as seismic (anomalous seismicity and micro-seismicity, swarms and fore-shocks, changes in b-value, hypocenter migration, changes in released energy and seismic waves velocities) and non-seismic (i.e. geophysical/geochemical precursors).

Geophysical precursors can be electromagnetic field variations, ground resistivity, telluric currents, ground deformation and crustal movements, tilt and strain and in earth tidal strain, water level changes. Meanwhile geochemical precursors are related with radon and other gases emissions, and chemical composition of underground water. These anomalous phenomena do not provide the basis for prediction of the three main parameters of an earthquake: place and time of occurrence and magnitude of the future seismic event but could forecast an increased probability of an imminent large earthquake occurrence. A more

detailed review of earthquake precursors can be found in Cicerone et al. (2009).

Since the 70s, when major efforts were made on promising precursors such as radon and $CO_2$ emissions, few advances have been achieved regarding earthquake precursors and how to develop a predictive model. However, on short time scales, less than a few months, earthquake sequences show a high degree of clustering in space and time. The probability of triggering increases with the initial shock's magnitude and decays with elapsed time according to simple scaling laws. The first generation

of models used for short-term clustering of earthquakes are the Omori law (Omori, 1984), Omori-Utsu law -also known as Modified-Omori (Utsu, 1961, 1969)-, and then Reasenberg and Jones (1989), in chronological order.

The Gutenberg-Richter (Equation 1, G-R from now on) empiric Law (Gutenberg and Ritcher, 1956) has been and still is one of the most used mathematical models that aims to explain the magnitude-frequency earthquake distribution based on the observations compiled in the catalogues. The main reason for its popularity is its simple formulation, which in turn enables a

more straightforward computer-based data process.

$$\log_{10} N_{M \geq m}(m) = a - b \cdot m \tag{1}$$

Where *a* is the earthquake productivity for the area; *b* is related to the ratio between high magnitude and low magnitude earthquakes and $N_{M \geq m}$ is the number of earthquakes with magnitudes *M* greater than a threshold magnitude, *m*.

The main constraint of this empiric law is that the catalogue must describe a homogeneous Poisson point process, that is,

the process is random and the average number of events that occur per time unit, $\lambda$, is constant (no matter the length of the considered interval).

Recent studies have shown the importance of the so-called b-value regarding seismic risk assessment by relating its low values (depending on the tectonic regime and the area) to tectonic stress build-up (Gulia and Wiemer, 2010) Moreover, the conclusions of this work agree with tests conducted in laboratory scale (Wiemer and Schorlemmer, 2007). Therefore, the

relationship demonstrated by De Santis et al. (2019) between b parameter and the Shannon Entropy has allowed the use of this thermodynamic variable as an indicator of the occurrence of an earthquake (Posadas et al., 2021, 2022); but, in addition, non-extensive entropy (Vallianatos et al., 2018; Vallianatos and Michas, 2020) is also likely to be used in the same terms (Papadakis et al., 2015). Finally, Galiana-Merino et al. (2022) proved the viability of using radon measurements to estimate

the daily seismic activity rate. Then, time-dependent seismic hazard or risk can be computed using seismic and non-seismic information (e.g., radon) to provide useful results for OEF.

All the previous studies rely on an accurate estimation of the spatiotemporal variations of the b-value. They will be discussed in the next part.

## 1.1 Temporal variations of the b-value

The temporal distribution of earthquakes for a given tectonic region is usually used to evaluate the change of the b-value before and after the occurrence of the main event in a certain area. The ability to monitor the seismic activity and the increase in the detection and characterization of the earthquakes in a certain area or fault is what makes this technique interesting.

The number of events used for the b-value estimation is a parameter that distinguishes two different methods: the rolling window method (RW) and the weighted likelihood method (WL).

RW has been used for a long time and can be seen in different studies (e.g., Gulia et al., 2016; Gulia and Wiemer, 2019; Smith, 1981). This method relies on the definition of an event window (considering a fixed number of events or for a certain period) for a stable b-value calculus. It is easy to implement and only requires a quick inspection of the temporal event distribution to determine the event window size.

Often, the size of this window is chosen arbitrarily which means there may be a better choice or that the results may be not accurate in some windows. DeSalvio and Rudolph (2021) pointed out in their work that the Traffic Light System developed by Gulia and Wiemer (2019) for forecasting earthquakes with fore-shocks higher than $M_w \geq 6$ should be evaluated with parameters obtained by optimization in the parameter space rather than handpicked values.

On the other hand, WL was introduced by Tormann et al. (2014) in order to eliminate the event window size choice (generally arbitrary) and has been used recently by Taroni et al. (2021a). It uses a time-decaying exponential weight function, which parameter has been optimized by means of an exponential likelihood function. The main advantage of this method is that it avoids any arbitrary choice of parameters, so using different datasets with the same algorithm is enabled without previous event distribution studies.

## 1.2 Spatial distribution of the b-value

The computation and mapping of the spatial distribution of the b-value is a useful tool regarding information showcasing. Depending on the data and the level of detail it can help describing structures such as faults or tectonic stress build up zones (García-Hernández et al., 2021).

There are several examples of b-value spatial mapping due to shallow seismicity. Tormann et al. (2014) mapped the details of several faults from California (U.S.) and their tectonic stress distribution by developing a distance-dependent sampling method. Taroni and Akinci (2021) included foreshocks and aftershocks in the computations by means of a weight function, so the catalogue does not need to be declustered, although all the existing seismic series have to be identified. Additionally, depth profiles of the b-value spatial distribution are also useful for analysing the tectonic behaviour responsible of the intermediate and deep seismicity. Amongst others, Wiemer and Wyss (2002) computed the b-value in-depth changes to understand the

seismic activity due to volcano related seismicity. More recently, Batte and Rümpker (2019) used this technique to analyse the shallow seismicity and relate the high b-values to heat flow in a rift environment. Chiba (2022) mapped the b-value distribution for the area from North Okinawa to Southern Kyushu Island (Japan), a region with a complex and rich tectonic setting, so the zones with more seismogenic potential are showcased.

When computing the spatial distribution of the b-value, different smoothing kernels are used to weight down the events depending on the distance from the spatial grid cell in which the b-value is calculated. The comparison between the event windows in the time series and the smoothing kernels can be made as they play the same role in the b-value calculus.

Another issue that has to be addressed is the method chosen for the estimation of the cut-off magnitude calculus. Recent work (Zhou et al., 2018) has shown that the characteristics of the seismic catalogue determine which algorithm suits better the cut-off or threshold magnitude calculus which is needed to calculate the b-value according to the maximum likelihood method proposed by Aki (1965) and improved by Utsu (1966).

Following the argument from the previous part regarding window sizes, the smoothing kernel should not be chosen arbitrarily, it will be necessary to find a way to correlate the event distribution with the smoothing kernel. Therefore, we propose to correlate the epicentre spatial distribution of the events in the catalogues to determine the smoothing kernel to be used in the seismic parameter spatial mapping. The function that best describes the spatial distribution of the events of the catalogue will be also the best to approach neighbouring events when calculating the b-value in a given spatial cell.

### 1.2.1 Smoothing kernels in spatial mapping

According to the definition of the weight functions, the parameter $\sigma$ (Brunsdon et al., 2002), regarded as a bandwidth, determines the focus or level of detail, so the higher this parameter, the slower this smoothing kernel function decays over space and vice versa. This means that the lower $\sigma$ values the higher level of detail.

Recently, Taroni et al. (2021b) refined Tormann et al. (2014) methodology to plot the spatial distribution of the b-value in Italy, and although there exists controversy in the conclusions (Gulia et al., 2022), our main interest is the methodology and the event catalogue. They employed the $\sigma$ value calculated by Murru et al. (2016), by means of the maximization of the likelihood of the seismicity contained in half of the Parametric Catalogue of the Historical Italian earthquakes (CPTI15 -Release v1.5-July 2016- from Rovida et al. (2020)) and obtained a smoothing parameter of 30 km for central Italy.

One of the main weaknesses of the non-Epidemic Type Aftershock Sequence (ETAS) b-value spatial mapping current methodologies is the fact that there is no clear justification on the use of the smoothing kernel function based on the event-spatial grid cell distance for regional b-value mapping. Both the spatial grid cell size and resolution are arbitrary and do not provide information of the seismic sources of the zone, as the grid's extent only purpose is to contain all the events of the catalogue. Although the influence of the grid choice can be minimized as pointed out by Tormann et al. (2014), this distance distribution does not relate to each event's source. We introduce a methodology that relies on the analysis of the event-event distance distribution for the fit of the smoothing kernel function and its parameters in order to optimize the resolution of b-value mapping.

## 2 Methodology

In order to obtain the smoothing kernel function and its parameters for a given seismic catalogue, we will follow the next steps.

First, it is necessary to study both the event-event distance and the spatial cell-event distance distribution. The event-event distance is the distance between any two events of the catalogue (in any of case studies the distances between all the event pairs will be calculated), as for the spatial-cell event distance, it is defined as the distance between a spatial grid cell and an event from the catalogue (as in the former definition the distances between all the spatial cells and all the events will be calculated). These quantities can be calculated once the Euclidean coordinates are obtained for each event and the spatial cell of the grid (Equation 2):

$$d = \sqrt{(x_1 - x_2)^2 + (y_1 - y_2)^2} \tag{2}$$

Where $d$ is the Euclidean distance, $x$ is the abscissa and $y$ is the ordinate of each point (event or centre of the spatial cell).

The main problem regarding this step arises when the catalogues are extensive. The number of events can increase the memory requirements for the event-event distance calculus, as for each event it will be necessary to calculate as many distances as events exist in the catalogue. For example, if the catalogue has 50000 events it will be necessary to allocate enough memory to store a 50000 by 50000 float-type array. This can also happen when calculating the spatial cell-event distance distribution depending on the area's size or the spatial cell resolution.

There are several ways to work around this situation. If the area does not display many seismic clusters, i.e., the distribution of the events can be approximated as a random one, then a set of random events in the catalogue can be used to calculate the distance distribution instead of the full catalogue. For this condition to be fulfilled, the area should be sized so no tectonic features (around which the events may cluster) have influence on the seismic record. Another option is to split the catalogue into several parts when the previous condition cannot be fulfilled. Tormann et al. (2014) proposed the introduction of a cut-off distance for the calculus of the spatial cell-event distance-based weight function values. According to their work, the events further than 7.5 km have no influence on the outcome so they can be ignored. To avoid the loss of resolution and stability they also include a minimum number of events for each spatial cell.

In our case, the same catalogue as Taroni et al. (2021b) will be used initially and all the events of the catalogue are considered for each spatial cell b-value calculus. Instead of using a cut-off distance, the influence of the events in the b-value calculus is controlled by means of the smoothing kernel and its parameters.

Once the distances have been calculated the next step is to plot these results and analyse them. If the distance distribution can be fitted to a function, then the smoothing kernel parameters will be obtained as a result of this fit. For example, if the distance distribution is identified as a normal distribution, then $\sigma$ will be calculated as the second moment of the distribution (the variance).

Lastly, the weight function that will control the influence of the events on the b-value calculus should be defined. In this case, it is the result of the product of two components: the fitted function that plays the role of smoothing kernel and then a

function that controls the weight of the events that belong to a seismic series following the procedures of Taroni and Akinci (2021). The final weight function can be defined as follows (Equation 3):

$$W = W_{SK} \cdot W_{SS} \tag{3}$$

Where $W_{SK}$ is the smoothing filter and $W_{SS}$ is the function used to add foreshocks and aftershocks into the b-value calculus. This weight function operates inside the expression of the b-value as defined by Utsu (1965) and adapted by Taroni et al. (2021b):

$$\hat{b} = \frac{1}{\left( \sum_{i=1}^{N} W_i \cdot (M_i - M_{min}) + \frac{\Delta}{2} \right) \cdot \log 10} \tag{4}$$

Where $N$ is the total number of events in the catalogue, $M$ is the magnitude of the event, $M_{min}$ is the threshold magnitude
and $\Delta$ is the binning of the magnitude in the catalogue. In these case studies the threshold magnitude does not change in a manner that can affect the b-value calculus, so no changes depending on time windows have been considered.

     The uncertainty of this b-value has been calculated following the procedure of Taroni et al. (2021b) and it was derived by these authors following Aki (1965) work and applying the delta method (Dorfman, 1938) to take into account the weight function used in the b-value calculation:

$$\hat{\sigma}_{\hat{b}} = \hat{b} \cdot \sqrt{\sum_{i=1}^{N} W_i^2} \tag{5}$$

The influence of the choice of the smoothing kernel is important, that is why it should be made by means of a function fit. Figure 1 depicts the difference between several functions used as smoothing kernel and the weight each event will be given depending on the distance (between the event and another event of the catalogue or the event and a given spatial cell):

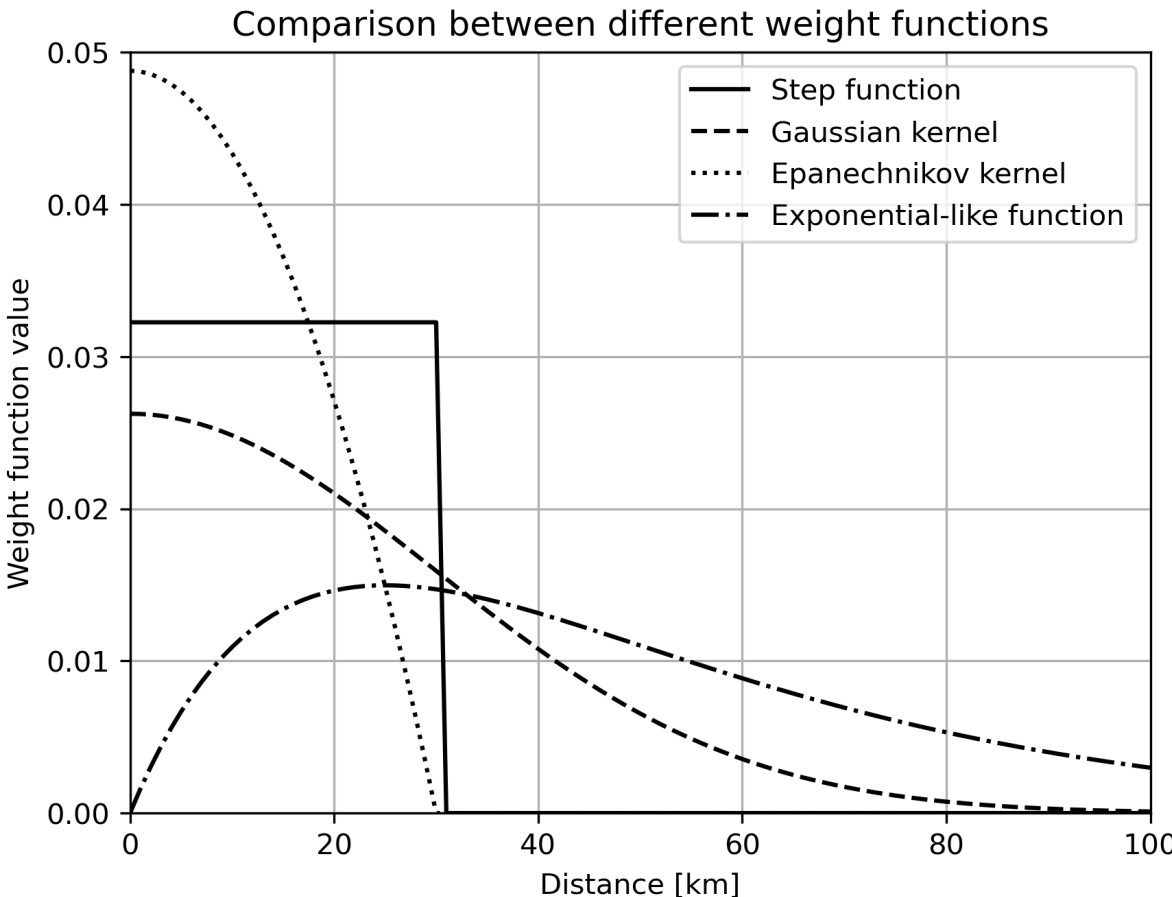

**Figure 1.** Comparison between different smoothing kernels and the weight the events are given towards the b-value calculus depending on the distance.

Our methodology will be first validated using the calibration catalogue for Italy (Taroni et al., 2021b) and then we will apply it to two different seismic environments: Lorca (southeast of Spain) in the time-span around the main earthquake in the last decades (the 2011 Lorca's Earthquake $M_w$ 5.1) and the Vrancea region (center of Romania), in a period of time also including two earthquakes of $M_w$ higher than 5. The Lorca region is dominated by shallow crustal earthquakes while the Vrancea region is mainly guided by intermediate and deep seismicity.

# 3 Results and discussion

## 3.1 Calibration catalogue: Italy

The Italian catalogue comprises the events from 1960 to 2019 for all Italy. It amounts up to 56309 events, which can be described in terms of magnitude and depth. The depth of the events ranges from 0 to 30.0 km, so the seismicity considered for this area is shallow (this catalogue has been filtered so no aftershocks or foreshocks nor events with depth greater than 30 km appear). As for the magnitudes, the minimum is 1.81 $M_w$ and the maximum is 6.81 $M_w$. All the events can be displayed in a frequency-magnitude plot in order to examine their distribution (Figure 2).

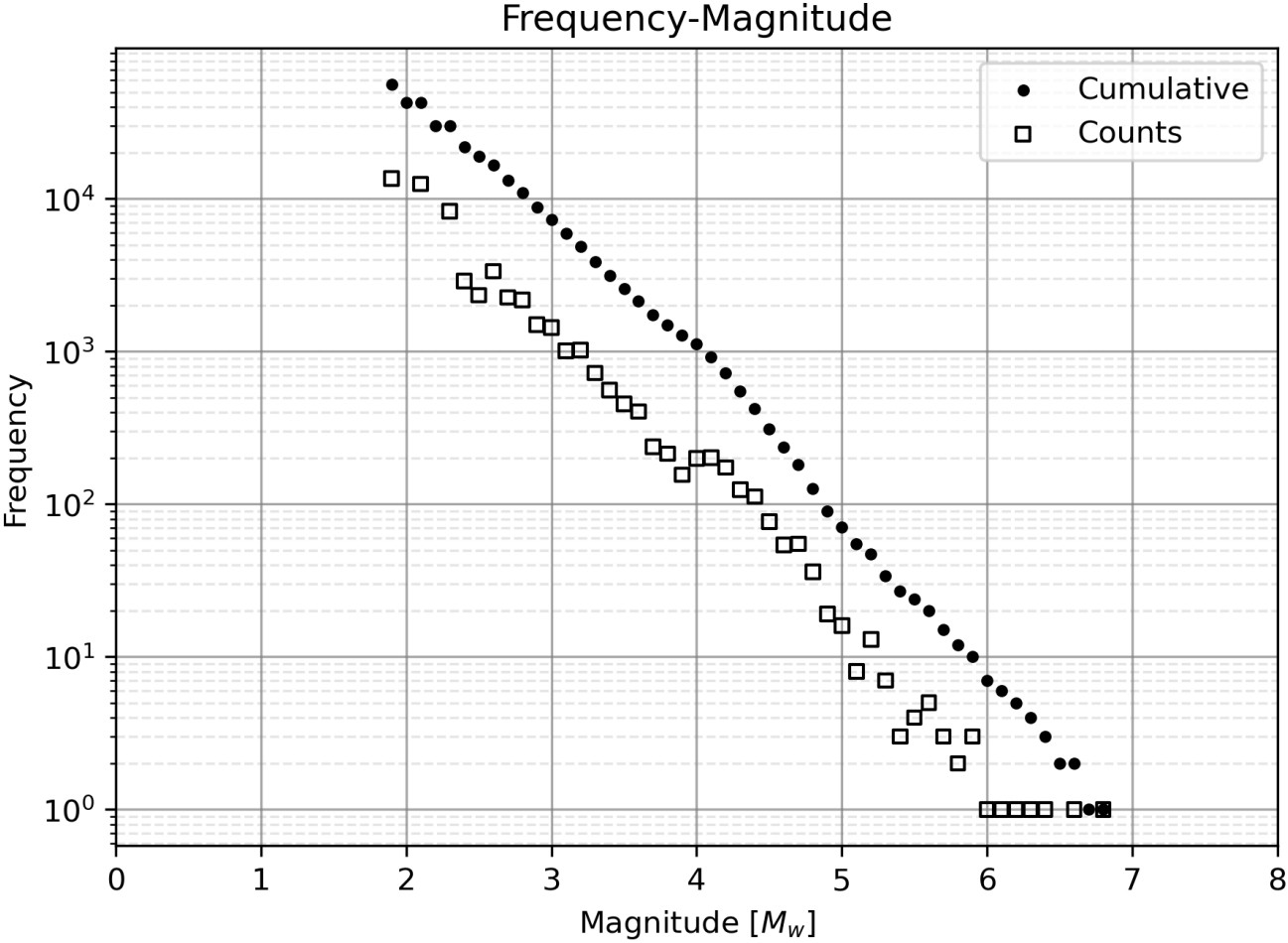

**Figure 2.** Frequency-magnitude plot for the Italian CPTI15 earthquake catalogue. This catalogue contains a total of 56309 events.

All the events in the catalogue have been used to plot the b-value map in order to compare the results with those obtained by Taroni et al. (2021b).

First, the distance between the $i_{th}$ event with the rest of the catalogue and between the $j_{th}$ spatial cell and the events of the catalogue ($\forall i, j$). These quantities will be called event-event distances and spatial cell-event distance from now on.

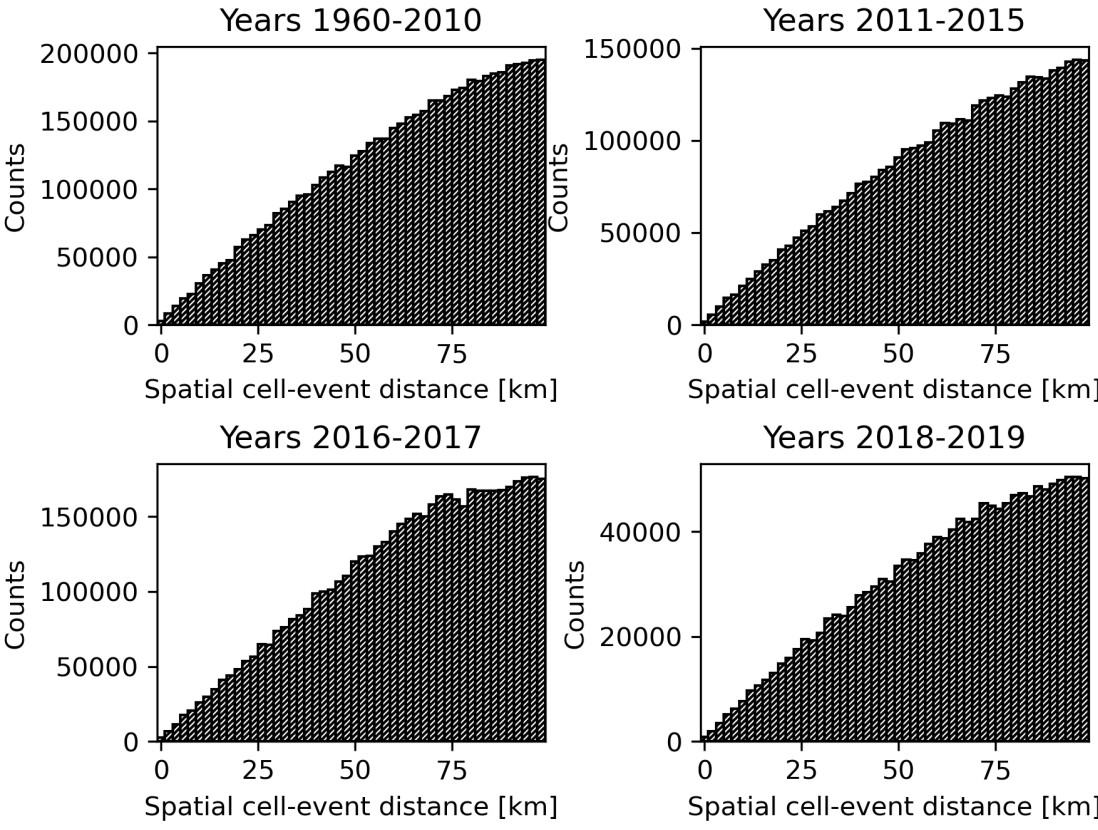

**Figure 3.** Histograms of the distances between every spatial cell and event pair (spatial cell-event distances) of the CPTI15 Italian earthquake catalogue at different time periods.

In Figure 3, it can be seen that the spatial cell-event distance does not give useful information since the distance distribution is the same for each period as it only depends in the shape of the grid and point density. For this reason, this quantity will not be considered in further case studies.

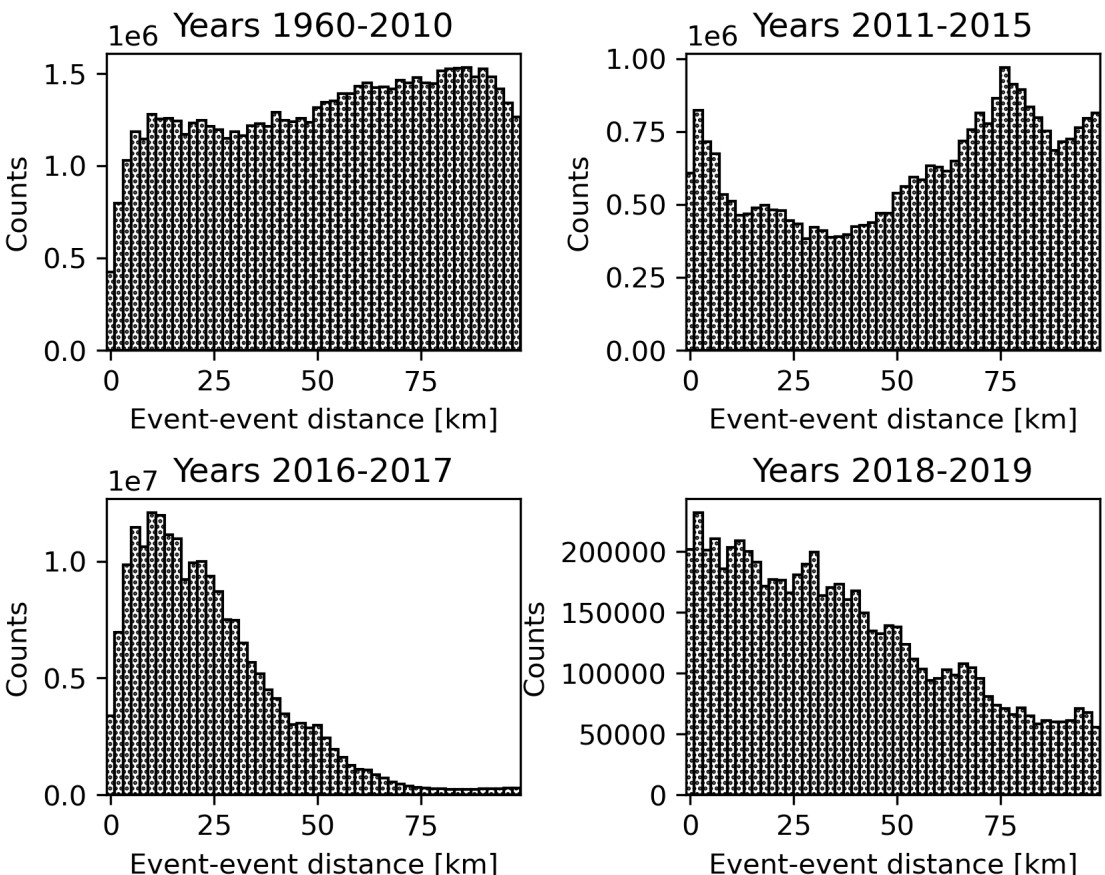

**Figure 4.** Histograms of the distances between every event pair (event-event distances) of the CPTI15 Italian earthquake catalogue at different time periods.

The event-event distance distribution shown in Figure 4 can be fitted to a different function depending on the period, but it is important to compare the number of counts in each histogram to draw further conclusions. For the period from 2016 to 2017 it can be seen that for each bin of the histogram the counts are one order of magnitude higher than for the rest of the periods. This means that this distance distribution will be representative of the catalogue except for the distances further than $50$ $\mathrm{km}$ (for which the counts are lower and the influence of the other histograms when summed can modify the distribution).

All the histograms have been stacked so the event-event distance distribution can be studied (Figure 5).

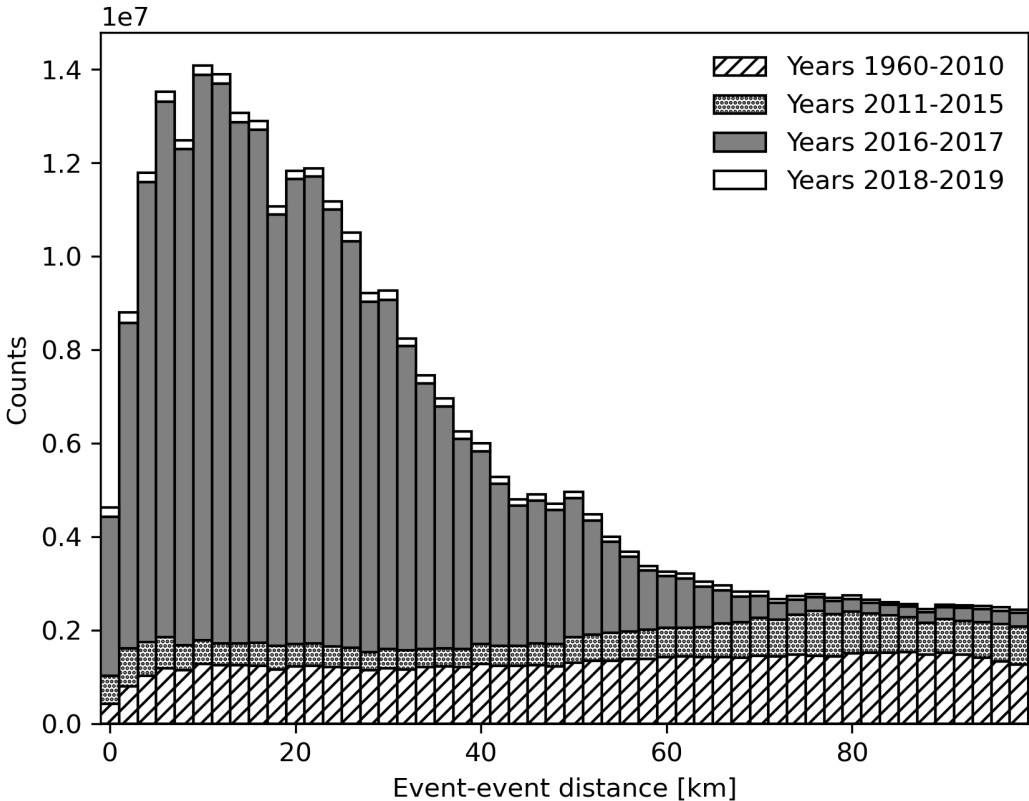

**Figure 5.** Stacked histogram with all the data from the event-event distances in the CPTI15 Italian earthquake catalogue.

In order to obtain the smoothing kernel, two functions have been considered based on existing literature for the exponential-like function (Tormann et al., 2014) and mathematical significance for the Gaussian kernel as this function has direct relationship with the distance distribution by means of the $\mu$ and $\sigma$ parameters. First, the Gaussian function (Equation 6):

$$f(r) = A \cdot \exp\left(-\frac{(r-\mu)^2}{2 \cdot \sigma^2}\right) \tag{6}$$

Where $A$ is the normalization constant, $\mu$, is the mean value or first moment of the distribution (the maximum value of the Gauss function) and $\sigma$ is the standard deviation or second moment of the distribution. These will be the parameters obtained by fitting the data to this model. Although, usually, the mean value is set to zero or the data is normalized to impose a zero-mean value for simplicity (Rasmussen and Williams, 2006).

An exponential-like function similar to the one used by Tormann et al. (2014) has also been considered (Equation 7):

$$f(r) = d \cdot r \cdot \exp(-r \cdot c) \tag{7}$$

Where $r$ is the distance and $c$ and $d$ are both real parameters that will be adjusted using the test data shown in Figure 5. For this function to be fitted, the count distribution has been normalized by dividing the counts of each bin by the sum of counts in all the bins so the parameters can be used for the weight function calculus without the need of a normalization constant.

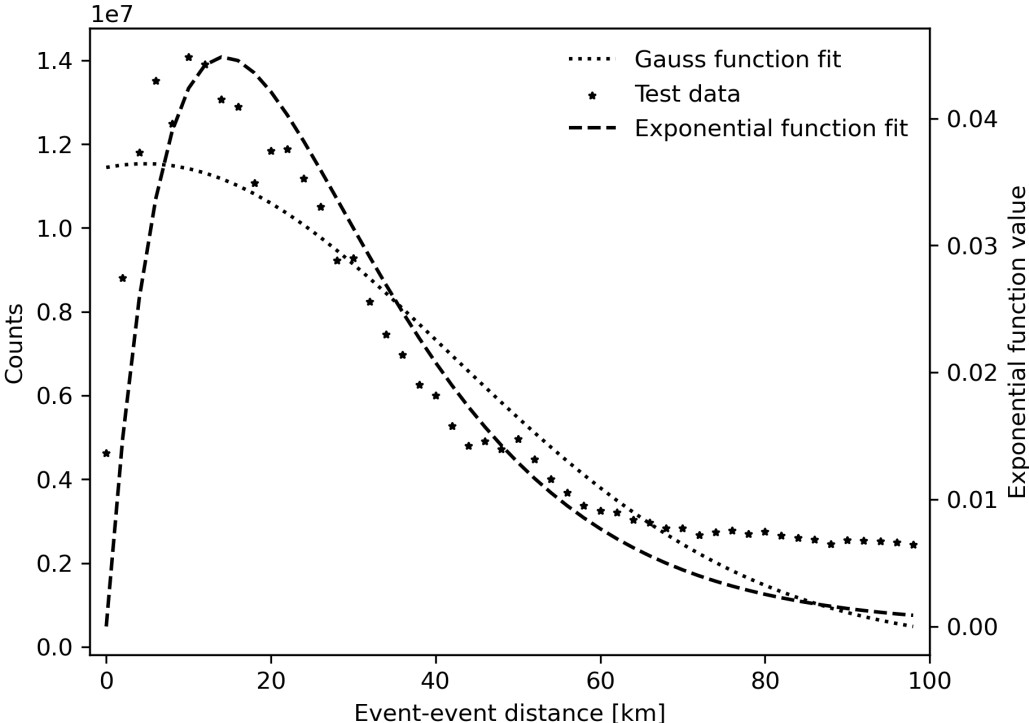

**Figure 6.** Comparison between both the Gaussian function fit and exponential-like function fit for the stacked counts in the event-event distance distribution for the Italian catalogue.

The exponential-like function is a better fit for the event-event distance distribution as it can be seen in both Figure 6 and Table 1, where the correlation coefficient - a measure of how much the points of the model function differ from those of the dataset - , $R^2$, is closer to 1 for the exponential-like function.

**Table 1.** Parameters obtained by fitting the Gaussian and exponential function to the counts in the event-event distance distribution for the Italian catalogue. Last column shows the $R^2$ of the model.

| Function | $\sigma / c$ | $\mu / d$ | $R^2$ |
|---|---|---|---|
| Gauss | $40 \pm 5$ | $5 \pm 7$ | 0.839 |
| Exponential | $0.070 \pm 0.003$ | $0.009 \pm 0.001$ | 0.918 |

The next step is the comparison of the results obtained by using the exponential-like kernel with those presented by Taroni et al. (2021b).

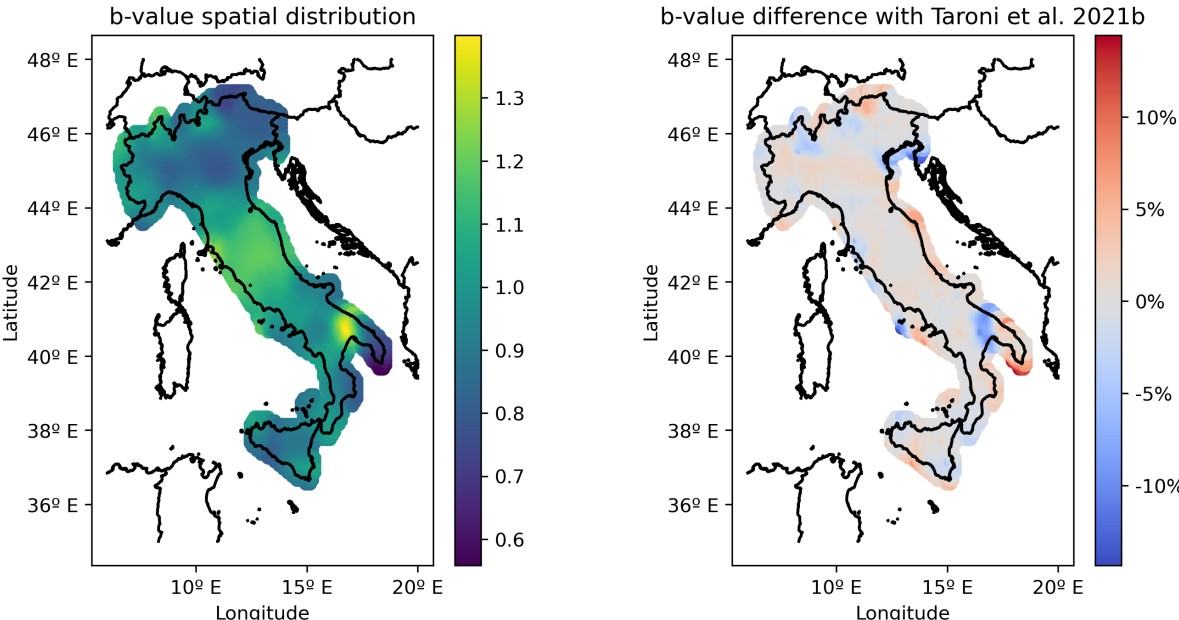

**Figure 7.** Spatial distribution of the b-values obtained in this work (left) and the percentage change of these with the b-values from Taroni et al. (2021b) (right). 56309 events have been used in the b-value calculus.

Only the spatial cell grids whose b-value is in the 95 % Confidence Interval (CI) with that of Taroni et al. (2021b) are plotted (only 10 spatial cells out of 4074 were outside of the 95 % CI). The difference between the two spatial maps is lower than 2 % in most of the country except for border areas in which the difference can rise up to a 15 % as it can be seen in Figure 7. This
can be due to less data being available for the b-value calculus (border effect).

Once the proposed methodology to obtain the smoothing kernel value in order to compute the b-value spatial distribution has been tested, it will be applied to two different case studies.

### 3.2 Case study: Lorca and Vrancea region

#### 3.2.1 Lorca's Area (Spain)

For the area of Lorca, we select the following region centered around the epicentre of Lorca's earthquake that happened the $11^{th}$ May 2011, 16:47 UTC. This event has been extensively studied. For instance, Martinez-Diaz et al. (2012) studied the rupture of the Alhama de Murcia fault and calculated the stress build-up and release using different fault models by means of interferometry data to account for the co-seismic deformation. González et al. (2012) studied the relationship between the crustal stress changes and the co-seismic slip distribution. Frontera et al. (2012) performed a comparison of the deformation

by means of data and numerical models. More information about Lorca's earthquake can be found in the special issue of the Bulletin of Earthquake Engineering (Alarcón and Benito, 2014).

In order to apply the proposed methodology, a part of the Spanish earthquake catalogue ( https://www.ign.es/web/ign/portal/sis-catalogo-terremotos) was filtered selecting the events in a 40 km radius circumference centered at Lorca's earthquake epicenter. Events have been selected from year 2000 to 2021 to have enough events to calculate the b-value (Figure 8). This

catalogue has a total of 2962 events with magnitudes between 0.8 $M_w$ and 5.0 $M_w$ (low to moderate earthquakes) and depths that range from 0 to 32.0 km (shallow seismicity). Before November 1997, epicentral location uncertainties were calculated with Hypo71 (Lee and Lahr, 1975) and specified as the so-called ERH (standard horizontal error, in km). However, since November 1997, epicentral location uncertainties calculated by Evloc (Carreño-Herrero and Valero-Zornoza, 2011) are reported as error ellipses at 90 % confidence level in the full-format catalogue. The epicentral location and the focal depth has

uncertainties usually lower than 5 km within the Iberian Peninsula (González, 2017).The threshold magnitude for shallow seismicity is $M_w$ 1.8.

## South-eastern Spain seismic catalogue (2000 - 2020)

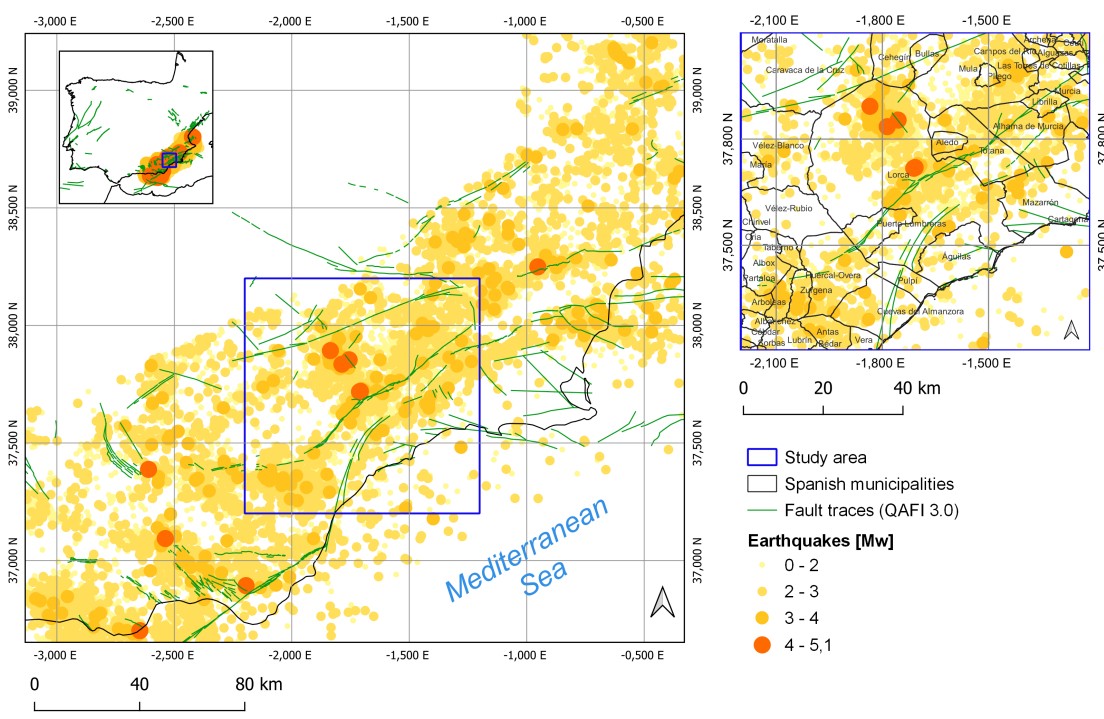

**Figure 8.** South-eastern Spain's seismic catalogue, zoom over Lorca's area (right) and fault traces in green from QAFI 3.0 (García-Mayordomo et al., 2012).

The catalogue has been divided in two parts: a pre-series period (from 2000 to 2011) and post-series period (from 2011 to 2020) so the b-value maps can be studied. Following the procedure described in the former example, the event-event distributions have been plotted and the two functions presented before being fitted in order to obtain the smoothing kernel function and its parameters. The clusters have been identified by means of the Reasenberg and Jones (1989) algorithm.

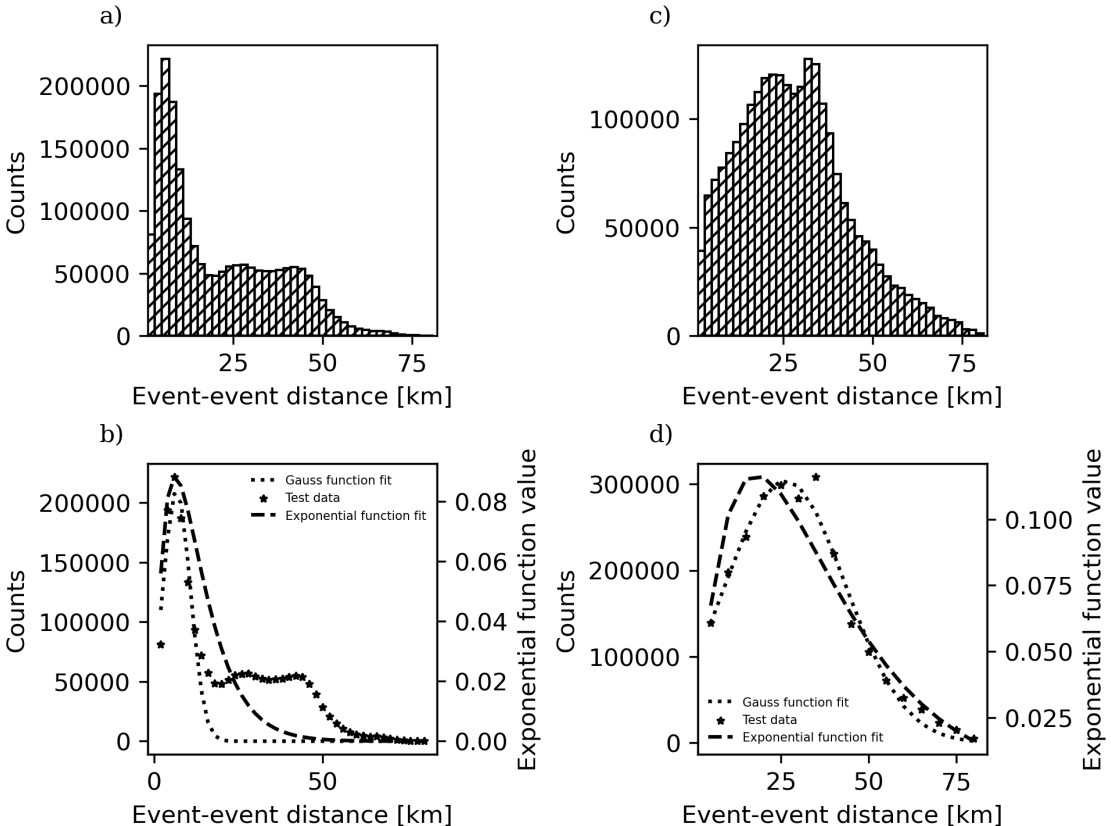

**Figure 9.** (a) Event-event distance distribution for events from 2000 to 2011. (b) Functions fit of the event-event distance distribution counts up to 14 km distance for the data from 2000 to 2011. (c) Event-event distance distribution for events from 2011 to 2020. (d) Functions fit of the event-event distance distribution counts up to 80 km distance for the data from 2011 to 2020.

Then, Figure 9 represents the event-event distance distribution. As it can be seen, there is no clear distribution for the $0 - 80$ km interval in the first period, a). However, the extension of the area allows considering only the first 14 km of the distance distribution. The Gaussian function has been fitted in b) with this constrain. As for the exponential function all the data has been used. For the second period (2011-2020), the event-event distance distribution in c) shows a clearer tendency, for this reason, the functions that have been fitted in d) use all the available data. The parameters obtained are represented in Table 2.

**Table 2.** Parameters obtained by fitting the Gaussian and exponential function to the counts in the event-event distance distribution for Lorca's area catalogue. Last column shows the $R^2$ of the model.

| Period [yrs] | $\sigma$ | $\mu$ | $c$ | $d$ | Gaussian $R^2$ | Exponential $R^2$ |
|---|---|---|---|---|---|---|
| 2000-2011 | $4 \pm 1$ | $6.7 \pm 0.4$ | $0.16 \pm 0.01$ | $0.04 \pm 0.01$ | 0.872 | 0.784 |
| 2011-2020 | $16 \pm 1$ | $26 \pm 1$ | $0.056 \pm 0.004$ | $0.018 \pm 0.002$ | 0.982 | 0.893 |

It can be seen in both the Figure 9 and the Table 2 that the Gaussian function is a better fit for the event-event distance distribution. Then, using this smoothing kernel, the b-value distribution for both periods is plotted:

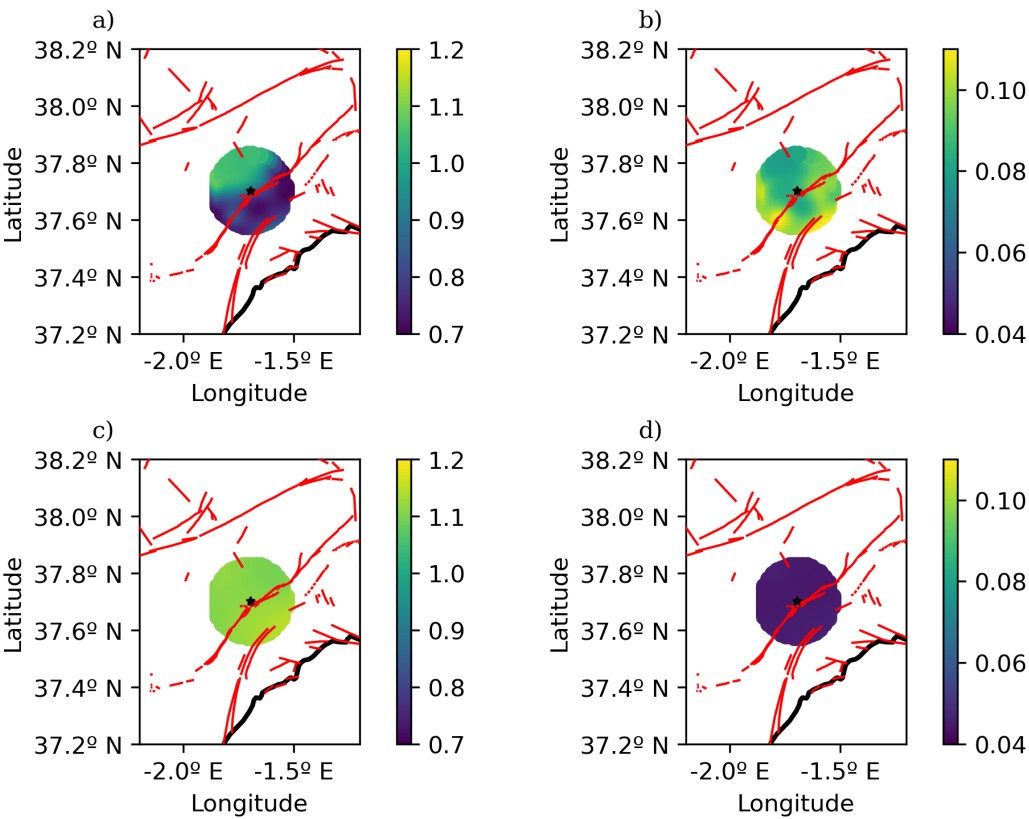

**Figure 10.** (a) Spatial distribution of the b-value and (b) its uncertainty in Lorca's area using data from 2000 to 2011 and the $\sigma$ and $\mu$ values obtained before. 981 events have been used in the b-value calculus. (c) Spatial distribution of the b-value and (d) its uncertainty in Lorca's area using data from 2011 to 2020 using the $\sigma$ and $\mu$ values obtained in the previous step. 824 events have been used in the b-value calculus. The red lines are the fault traces from QAFI 3.0 (García-Mayordomo et al., 2012) and the black star marker shows the location of Lorca's earthquake epicenter.

Figure 10a shows that the b-values in the proximity to the fault responsible of the Lorca earthquake (in the center of the circle) are lower than the tectonic zone's mean value (b = 1.03, García-Mayordomo et al., 2012). As this part of the catalogue comprises ten years before the Lorca's earthquake, the b-value spatial distribution calculated shows a zone of increased tectonic stress in the NE and SW parts of the fault, before the main Lorca's series event.

In Figure 10c it can be seen that the b-values are higher than the average value cited before. This could imply that part of the tectonic stress build-up has been released by means of the earthquakes of Lorca's series.

### 3.2.2 Vrancea region (Romania)

The Vrancea region, 135 km apart from Romania's capital, Bucharest, is one of the zones with highest seismic activity of Europe. Historical records of earthquakes show evidence of events with $M_s$ higher than 7 (the earthquakes from 1802 and 1838) and recent events (1990's) have reached $M_w$ higher than 6 (Zaicenco et al., 2008).

It is noteworthy to highlight that all strong earthquakes registered have occurred at depths greater than 60 km, which have become the focus of research for this zone. The up-to-date catalogue of Romania can be found in the following address: http://www.infp.ro/index.php?i=romplus. Between 1990 and the end of 2013, locations were determined using the HYPOPLUS (Oncescu et al., 1996) program, a 1D velocity model and stations corrections. Starting with 2014, the earthquake location is obtained using Antelope software. In the present form, a single magnitude scale (Mw moment magnitude scale) is adopted for all the events. Different magnitude scales used before 2014 were converted into moment magnitude (Mw), based on calibration relations presented in Oncescu et al. (1999) work.

# Vrancea County- Romania

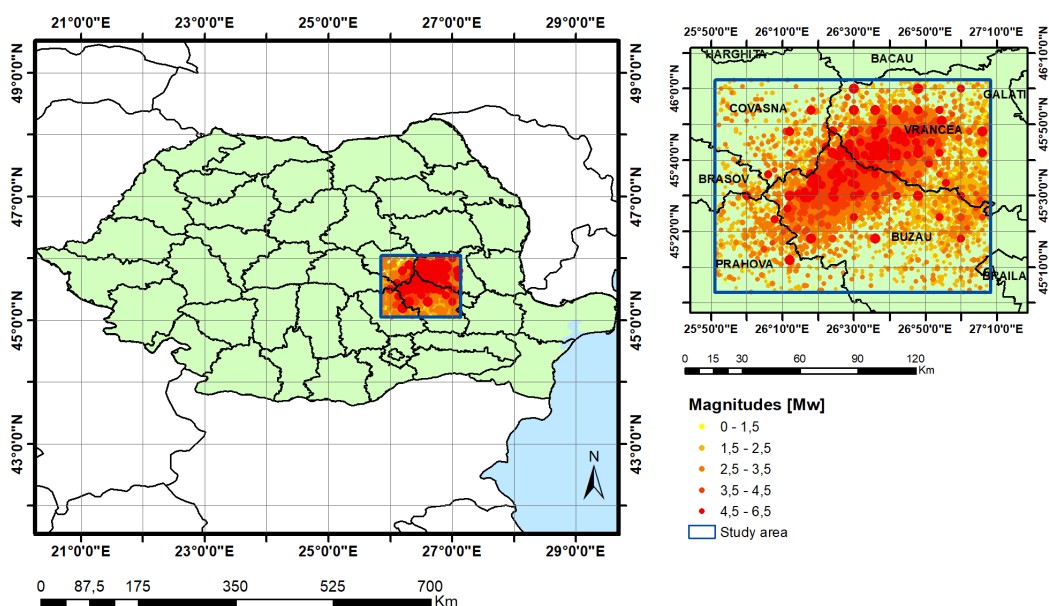

**Figure 11.** Plot of the catalogue from 2000 to 2018 in the Vrancea region (Romania). Romanian province shapefile obtained from WorldBank: Counties of Romania (National Agency of Cadastre and Land Registration).

Taking into consideration that the region has suffered two earthquakes with $M_w$ 5.5 in the last decade, we have chosen the events from 2000 to 2018 in the Vrancea region. The b-value mapping area is comprised inside the blue frame in Figure 11, in which all the events of the catalogue for this period are shown.

    The catalogue contains 6615 events with magnitude ranging from $M_w$ 0.1 to $M_w$ 6.0, 35 % have shallow depth, 19 % have intermediate depth and 46 % have deep depth. As we can see the most of them (65 %) are intermediate and deep seismicity.

The cut-off magnitude for this catalogue is $M_w$ 2.70.

    The events can be plotted in the frequency-magnitude graph depending on their depth. In this case deep seismicity accounts for intermediate and deep events (> 50 km depth) and shallow seismicity for those with depth lesser than 50 km. In this case, the catalogue has not been filtered prior calculus (the Italian and Spanish only contain shallow seismicity; hence they do not require a G-R fit to discriminate the different seismic settings).

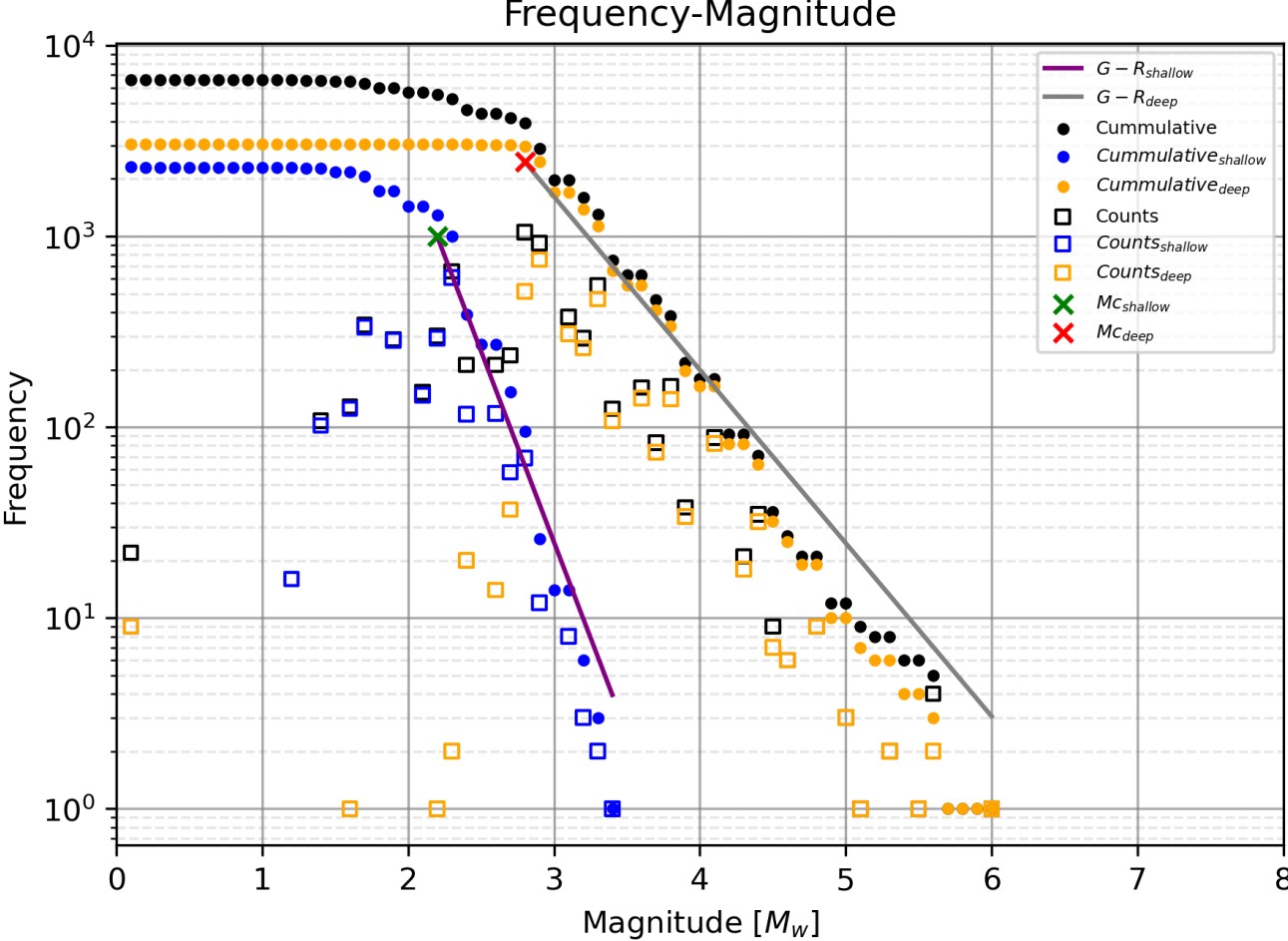

**Figure 12.** Frequency-Magnitude plot for the full catalogue (black), shallow events (< 50 km depth) and deep events (> 50 km). The counts are shown with void square markers and the cumulative representation in dots. The G-R law has been fitted for both the deep events and shallow events catalogue.

Figure 12 represents the magnitude-frequency distribution of the earthquakes at different depths and the tendency of the G-R law. A b-value of 2.00 for the shallow seismicity has been found and a b-value of 0.91 has been calculated for the deep seismicity. The different slopes indicate that separate catalogues should have been used to compute the spatial b-value distribution for each depth range. As the moderate to large earthquakes have a depth greater than 50 km, then we will consider only intermediate and deep seismicity for the study.

In this zone two major earthquakes occurred: the September $24^{th}$ of September 2016, 23:11 UTC and the $28^{th}$ of October 2018, 00:38 UTC . The catalogue will be split in two parts: from 2000 to 2016, and from 2016 to 2018 (both periods before

the respective earthquakes happened) in order to compare the b-value spatial distribution. The first step is to calculate the event-event distance distribution as in the previous cases (Figure 13).

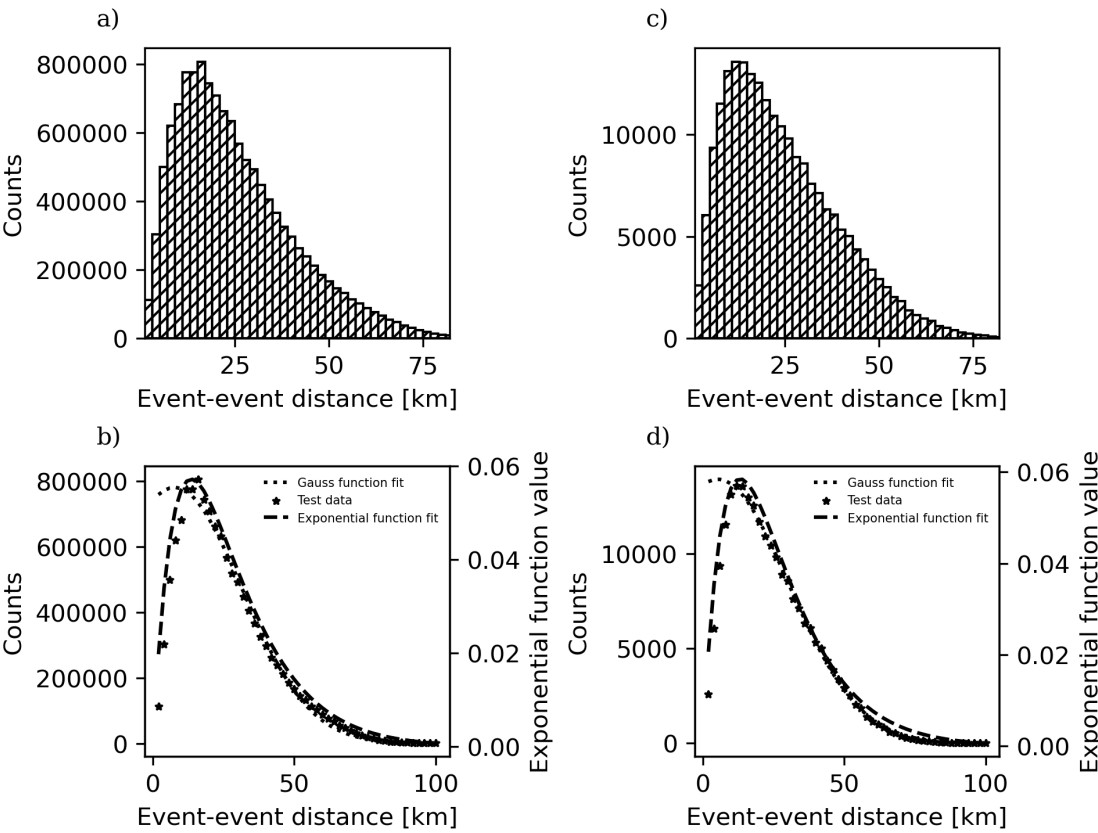

**Figure 13.** (a) Event-event distance distribution for events from 2000 to 2016. (b) Functions fit for the event-event distance distribution for events from 2000 to 2016. (c) Event-event distance distribution for the events from 2016 to 2018. (d) Functions fit for the event-event distance distribution for events from 2016 to 2018.

Both of the functions seem to fit the data, although the Gauss function overestimates the distribution in the first kilometers
and the exponential function overestimates the last kilometers. The parameters obtained by means of the functions fit are provided in Table 3:

**Table 3.** Parameters obtained by fitting the Gaussian and exponential function to the counts in the event-event distance distribution for the Vrancea region catalogue. Last column shows the $R^2$ of the model.

| Period [yrs] | $\sigma$ | $\mu$ | c | d | Gaussian $R^2$ | Exponential $R^2$ |
|---|---|---|---|---|---|---|
| 2000-2016 | $24 \pm 1$ | $8 \pm 2$ | $0.074 \pm 0.001$ | $0.0115 \pm 0.0003$ | 0.993 | 0.985 |
| 2016-2018 | $25 \pm 1$ | $6 \pm 1$ | $0.075 \pm 0.001$ | $0.0120 \pm 0.0003$ | 0.998 | 0.992 |

Both functions will be compared. First, the Gaussian function will be used as smoothing kernel (Figure 14).

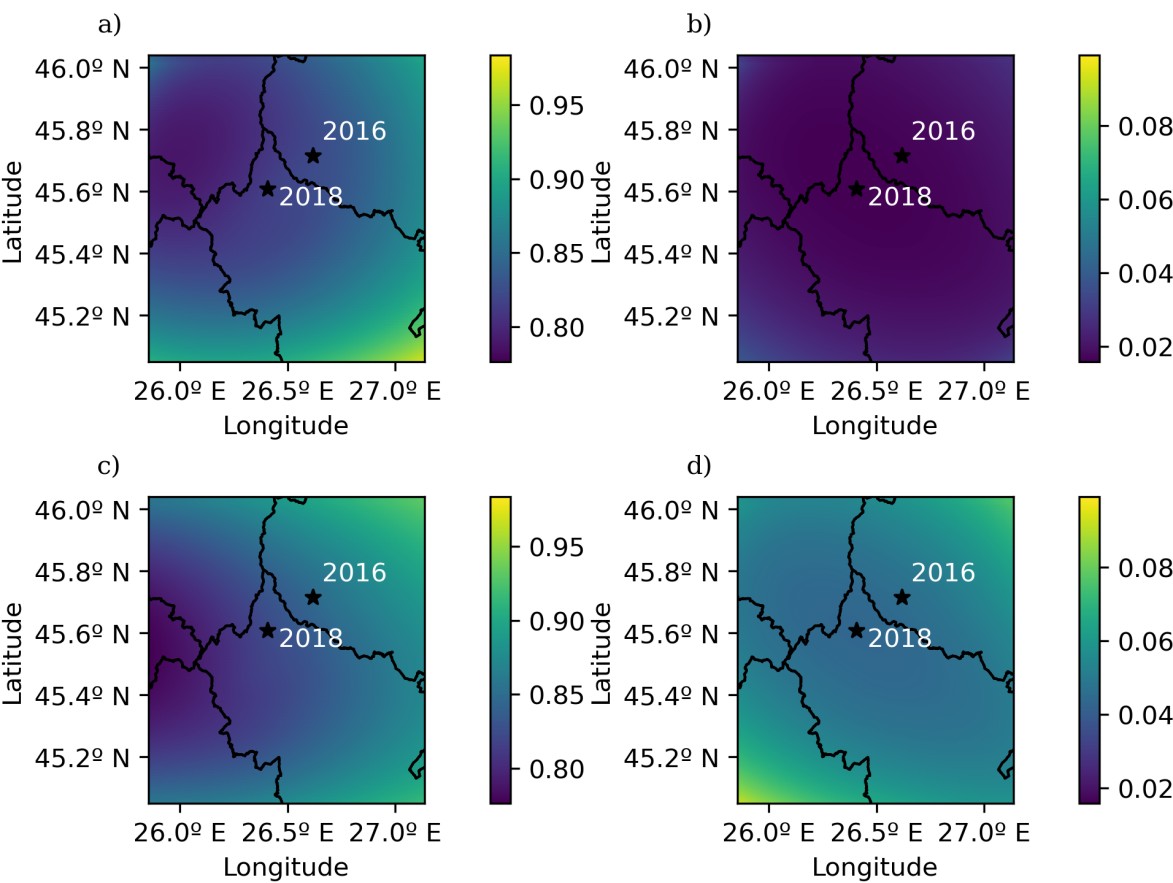

**Figure 14.** (a) b-value spatial mapping for the Vrancea region from 2000 to 2016 using the parameters calculated in the previous Gaussian function fit. (b) b-value uncertainty for the Vrancea region from 2000 to 2016. (c) b-value spatial mapping for the catalogue of the Vrancea region from 2016 to 2018 using the Gaussian kernel. (d) b-value uncertainty for the Vrancea region from 2016 to 2018. The black star in (a) and (b) marks the 2016 earthquake, and in (c) and (d) the 2018 earthquake.

As it can be seen in Figure 13b and c the fit of the Gaussian function in the first 10 km is not optimal. Therefore, the exponential-like function -which fits better at the first kilometers of the event-event distance distribution- can be used in order to compare the results (Figure 15).

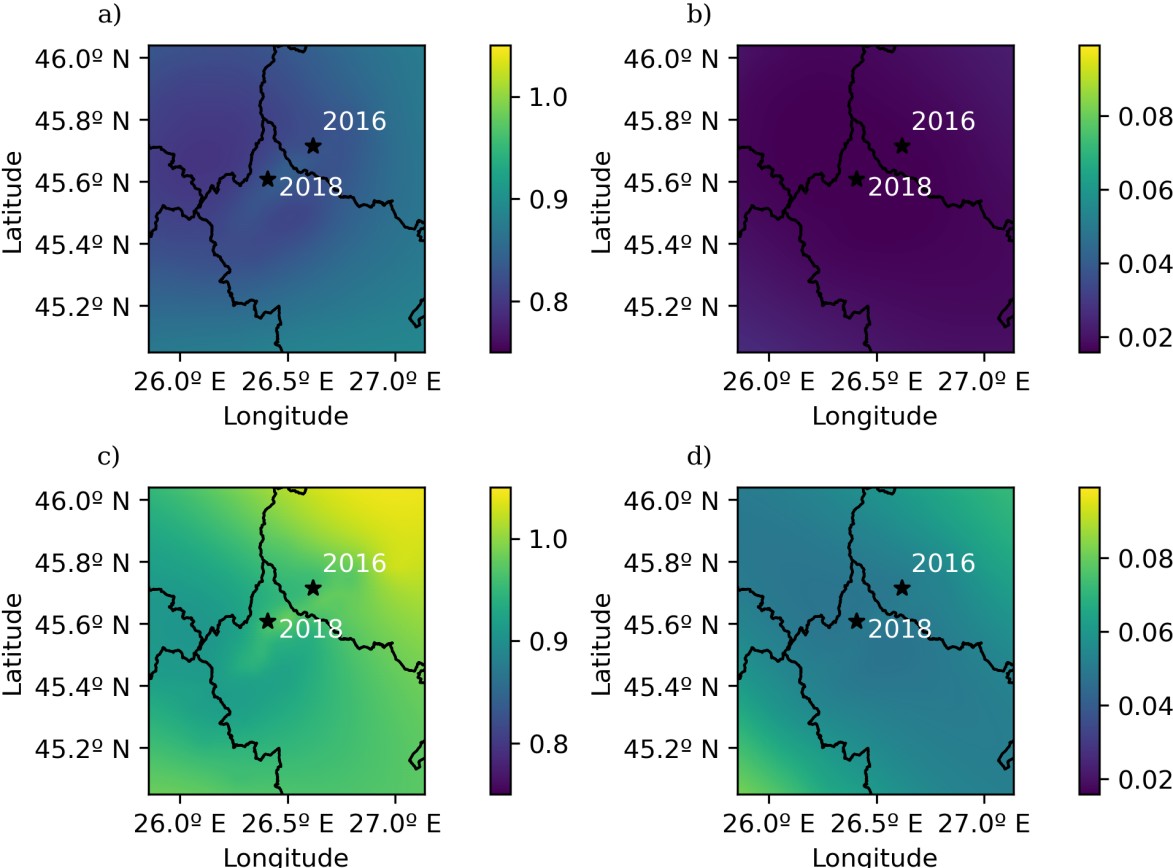

**Figure 15.** (a) b-value spatial mapping for the Vrancea region from 2000 to 2016 using the parameters calculated in the previous exponential function fit. (b) b-value uncertainty for the Vrancea region from 2000 to 2016. (c) b-value spatial mapping for the catalogue of the Vrancea region from 2016 to 2018 using the exponential kernel. (d) b-value uncertainty for the Vrancea region from 2016 to 2018. The black star in (a) and (b) marks the 2016 earthquake, and in (c) and (d) the 2018 earthquake.

The b-value maps in Figure 14c and Figure 15c when compared with those in Figure 14a and Figure 15a allow to identity an increase in the b-value near the epicentre of the earthquake of 2016 which could indicate tectonic stress relief and a slight b-value decrease towards the SW part of the area. This shifting could account to the tectonic stress build-up preceding the 2018 earthquake. The main difference between the exponential function and the Gauss function when used as spatial kernels in this case is the level of detail and the b-value range (difference between the maximum and minimum value). This can be explained by examining the graphs of the exponential function fit in Figure 13b and d. The decay in the exponential function is faster

than the Gauss function, so the influence of weight function in the b-value calculus is lesser for the most remote events in the exponential function than it is in the Gaussian function.

## 4 Conclusions

This method for the smoothing kernel assessment and the calculus of its parameters is able to obtain results compatible with those obtained by different methods (likelihood function). Moreover, it avoids the arbitrary selection of a smoothing kernel by fitting a function to the distance distribution and obtaining the parameters for it.

Although the spatial cell-event distance histogram has been shown for comparison purposes, it serves no use to the spatial kernel function calculus as the spatial grid is arbitrary (both in its extension and resolution) and the only constraint for this grid

is to cover the entire area in which the events of the catalogue are located.

It is illustrative to use the Gaussian function as the fit for the event-event distribution as it is a well-known function with parameters that can be easily related to the event-event distance distribution and its characteristics. Nevertheless, other distributions can be considered as long as the function that describes them is compatible with the data as shown in the Vrancea region case study.

Another interesting topic that can be addressed is the relationship between the $\sigma$ value in the case of the Gaussian function (as it is directly related with the event distribution) and the fault distribution in the study area, in case there exists one, as it could enable tectonic structure profiling.

The use of different parts of the catalogue in order to describe the b-value spatial distribution as the time passes can enable OEF as long as there is enough data for it to be stable. We used parts of the catalogue before and after a major earthquake, but

it could be used to describe yearly (or even monthly depending on the zone) b-value changes.

*Code and data availability.* The data and code used in this study is available upon request.

*Author contributions.* DML and SM developed the original idea. JJGM and IGD helped in the development of the methodology. DML and SM performed the data curation. DML wrote the code and tested the different components. All the co-authors participated in the investigation and result interpretation. DML prepared the original draft, and SM, JJGM and IGD reviewed and edited it.

*Competing interests.* The authors declare that they have no conflict of interest.

*Acknowledgements.* This study was supported by the European Union's Horizon 2020 research and innovation program under grant agreement Nº 821046, the Spanish Government through research project PID2021-123135OB-C21, and Research Group VIGROB-116 (University of Alicante).

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
