# Peer review of "On the calculus of smoothing kernels for seismic parameter spatial mapping: methodology and examples"

_Natural Hazards and Earth System Sciences, 2022_

## Referee Comment (RC1)

MS No.: nhess-2022-191

**Title: On the calculus of smoothing kernels for seismic parameter spatial mapping: methodology and examples**

**Author(s): David Montiel-López, Sergio Molina, Juan José Galiana-Merino, and Igor Gómez**

**MS type: Research article**

This paper presents a new method to display information about seismic parameters of a certain area by using smoothing kernels, that is, the spatial mapping of relevant information to enable operational earthquake forecasting (OEF) and, also, to define the tectonic source profiling. The scope of Natural Hazards and Earth System Sciences (NHESS) covers earthquake hazards and, in fact, some of the aims of the journal are (1) the study of the evolution of natural systems towards extreme conditions, and the detection and monitoring of precursors of the evolution, (2) the detection, monitoring, and modelling of natural phenomena, and the integration of measurements and models for the understanding and forecasting of the behaviour and the spatial and temporal evolution of hazardous natural events as well as their consequences and (3) the design, development, experimentation, and validation of new techniques, methods, and tools for the detection, mapping, monitoring, and modelling of natural hazards and their human, environmental, and societal consequences. Without a shadow of a doubt, the topic is suitable for Natural Hazards and Earth System Sciences and it is of broad international interest. Due to I am not a native English speaker, I have not any recommendation about English grammar or use of this language.

Basically, the proposed method replaces the spatial cell-event distance by the inter-event distances to built-up the kernel function; this has the advantage that while in the former it is necessary to take an arbitrary distance (7.5 km) from which the events are regard as to be uncorrelated, in the latter this is not necessary. In order to obtain the smoothing kernel, two functions have been considered as fit for the data: the Gaussian function and an exponential-like function and after fitting data, authors concluded that the second one is better than the first one. Then, they calibrated the function by using a previous study in Italy earthquakes catalogue (with respect to Taroni et al, 2021 paper) and finally, they applied the method to Lorca (Spain) earthquakes and to Vancrea (Romania) earthquakes.

In my opinion, this study on the whole is up to international standards and the assumptions to apply the new methodology from authors are reasonable. Moreover, I think that several meaningful conclusions are reached from results. I strongly believe that the revised manuscript has an excellent scientific quality and, of course, there is not misconceptions.

However, I am concern about some aspects and statements in the paper and I really recommend authors to incorporate them in order to clarify the final manuscript.

1. To my point of view, a more detailed description of the seismicity that is being studied and also, the fundamental parameters of the used catalogue, are missing. For example, maximum and minimum magnitude of the catalogue, threshold magnitude, generic errors

in the hypocentres locations (and therefore, time sensitivity of the located hypocentres), etc.

2. Although throughout the paper the calculation of parameter b is frequently discussed, at no time is it indicated which method has been used for this purpose. The only reference is (line 91 to 93):

> *"Another issue that has to be addressed is the method chosen for the estimation of the cut-off magnitude calculus. Recent work (Zhou et al., 2018) has shown that the characteristics of the seismic catalogue determine which algorithm suits better the cut-off or threshold magnitude calculus which is needed to calculate the b-value according to Aki (1965)."*

from which it is assumed that the classic formula of Aki (1965) has been used. It should be clarified and if this has been the expression used, the authors should explain why the common improvement made from Utsu (1966) has not been used.

3. Authors state that the exponential type function fits to data better than the Gaussian (lines 183 and 184). It should be justified using the correlation coefficient from table 1, such as with table two they did.

4. Abbreviation CPTI15 is not explained (line 106).

5. Although the number and quality of the references appropriate and they are accessible by scientists, there exist other significant advances (that should be added) and it bring this manuscript in accord with the recent literature; for instance, (line 47 to 50) can be re-written as follow:

> *"Recent studies have shown the importance of the so-called b-value regarding seismic risk assessment by relating its low values (depending on the tectonic regime and the area) to tectonic stress build-up (Gulia and Wiemer, 2010) Moreover, the conclusions of this work agree with tests conducted in laboratory scale (Wiemer and Schorlemmer, 2007).* **Therefore, the relationship demonstrated by De Santis et al. (2019) between b parameter and the Shannon Entropy has allowed the use of this thermodynamic variable as an indicator of the occurrence of an earthquake (Posadas et al., 2021, 2022); but, in addition, non-extensive entropy (Vallianatos et al. 2018, 2020) is also likely to be used in the same terms (Papadakis et al., 2015).** *Finally, Galiana-50 Merino et al. (2022) proved the viability of using radon measurements to estimate the daily seismic activity rate."*

> *References:*

> *De Santis, A., Abbattista, C., Alfonsi, L., Amoruso, L., Campuzano, S., Carbone, M., Cesaroni, C., Cianchini, G., De Franceschi, G., De Santis, A., Di Giovambattista, R., Marchetti, D., Martino, L., Perrone, L., Piscini, A., Rainone, M., Soldani, M., Spogli, L., Santoro, F., "Geosystemics View of Earthquakes", Entropy 21, 412-442 (2019).*

*Papadakis G, Vallianatos F, Sammonds P. A nonextensive statistical physics analysis of the 1995 Kobe, Japan earthquake. Pure ApplGeophys, 2015; 172:1923–31. https://doi.org/10.1007/s00024-014-0876-x.*

*Posadas, A., Morales, J., Ibáñez, J., Posadas-Garzon, A., Shaking earth: Non-linear seismic processes and the second law of thermodynamics: A case study from Canterbury (New Zealand) earthquakes, Chaos, Solitons and Fractals. Nonlinear Science, and Nonequilibrium and Complex Phenomena, 151, 2022. doi.org/10.1016/j.chaos.2021.111243.*

*Posadas, A., Morales, J., Posadas-Garzon, A., Earthquakes and entropy: Characterization of occurrence of earthquakes in southern Spain and Alboran Sea, Chaos: An Interdisciplinary Journal of Nonlinear Science,, 31 (4), 2021. doi.org/10.1063/5.0031844.*

*Vallianatos, F., Michas, G., Papadakis., G., Nonextensive Statistical Seismology: An Overview. Complexity of Seismic Time Series. In Chelidze, T., Vallianatos, F., Telesca, L., editors. Complexity of Seismic Time Series: Measurement and Application. Elsevier, 2018, p25-59. https://doi.org/10.1016/B978-0-12-813138-1.00002-X*

*Vallianatos, F., Michas, G., Complexity of Fracturing in Terms of Non-Extensive Statistical Physics: From Earthquake Faults to Arctic Sea Ice Fracturing, Entropy 2020, 22, 1194; doi:10.3390/e22111194.*

6. Figure and table captions are too short. One should be able to fully understand the meaning of the figure or table without appealing to the body of the manuscript. For example, in table 1 caption it is not explained what is R and what is S (nor in the main text) or, in figures 2 and 3, definition of spatial cell-event distance and the inter-event distances should be indicated. On the whole, a more detailed, broad and comprehensive captions are needed.

Finally, in my view, the paper is concise and properly organized (introduction, methodology, results and discussion and, at the end, conclusions). Reading only the title, I can easily imagine what the aim of the paper is and then, reading the abstract, I can get enough information about methods and conclusions of the manuscript. Concerning with data, they are public and they are available from corresponding author upon request. In addition, conclusions are directly supported by data.

In short, the present manuscript is worthwhile of publication in Natural Hazards and Earth System Sciences provided that the aforementioned points will be appropriately addressed by the authors.

---

## Referee Comment (RC2)

**Reviewer Comments**

The paper "On the calculus of smoothing kernels for seismic parameter spatial mapping: methodology and examples" by Montiel-López et al. presents a new method for the selection of smoothing kernels, based on the inter-event distance distribution between successive earthquakes, for the spatial mapping of seismic b-values. The adequate mapping of b-values in seismically active regions and their variations in time and space can contribute deciphering the state of stress and making inferences regarding seismic hazard. The main aim of the paper is to contribute to this field and clearly falls within the scope of Natural Hazards and Earth System Sciences. The paper introduces the topic in a consistent way, is generally well-written and structured, but there are some points that seem rather vague and require further clarification (see comments below). Therefore, I recommend revisions for the paper before it can be further considered for publication.

1) Inter-event distance is usually a term describing the distance between successive earthquakes. In the paper, it seems that this term is used to describe the distance between each event and all the other events in the catalogue. This point needs some further clarification in the part where Equation 2 is given. Apparently, only the coordinates of the epicenters are used to estimate inter-event distances. Is there a reason why depths are not used in the calculations? Please explain.

2) How exactly the weight function (Equation 3) works in the calculations? The authors cite the paper of Taroni and Akinci (2021), but some further clarifications and examples are needed to inform the reader how the weight function affects the b-value estimations.

3) Provide some more information regarding the Italian seismic catalogue (source etc.).

4) In Figure 3, the inter-event distance histograms are shown for different time periods. How were these time periods chosen by the authors? For instance, we see a 50-years interval followed by a 4-years and 1-year interval.

5) The authors consider two functions, the Gaussian and exponential functions, to obtain the smoothing kernel. However, power-law functions (e.g., Abe and Suzuki, 2003 and Corral, 2006) have also been considered in the literature, with their significance on the spatial clustering of earthquakes. Did the authors check the adequacy of such functions? A histogram in log-log axes will assist to show if power-law regimes exist.

6) In line 181, the authors say, "For this function to be fitted, the count distribution has been normalized". How exactly it is normalized?

7) With which method exactly are the b-values and their uncertainties calculated?

8) Is the difference between the b-value spatial distribution for Italy obtained in this work (Figure 6) within the confidence limits with that of Taroni et al.? The authors provide only the percentage change.

9) For the Lorca case, are the coordinates in Figure 7 and 9 the same? In the first figure we see positive values and in the second negative values for Longitude. In addition, the authors say that they selected the events at 40 km radius circumference centered at Lorca's earthquake epicenter. However, in Figure 7 we only see a fraction in a much smaller area. The text and the figures should be consistent.

10) Why do the authors show the 1579 – 2021 seismicity in Figure 7 after all? Only 2000 – 2021 seismicity is studied.

11) Why is year 2011 excluded from the analysis for the Lorca case?

12) In the caption of Figure 11 in c), what is "down" stands for?

13) In Line 210, the 0 - 100 km distance range is mentioned, but the radius for the seismicity selection is 40 km. Please be precise.

14) In Line 214 correct date with data.

15) The use of the words "in each spatial cell" in the caption of Figure 9 adds some confusion. Do the authors mean the total number of events used in the analysis?

16) In Line 230, the authors say, "Taking into consideration that the region has suffered three earthquakes with Mw 5.5 in the last decade, then we have chosen the events from 2016 to 2020 in the Vrancea region." This sentence does not make much sense and the selection is not justified. Probably the sentence needs rephrasing.

17) Add the b-values in Figure 11 or discuss them in the text. Why not similar plots are shown for the Lorca case, or the G-R fitting for the Italian catalogue?

18) Again, the authors exclude year 2016 for Vrancea. Is this a form of "manual declustering" excluding major aftershock sequences?

19) Rephrase the sentence "Both b-value maps (with the different kernels) depict an increase of the b-value in zone where the the earthquake of 2016 (Figure 13c and Figure 14c) which could indicate tectonic stress relief and a slight b-value decrease towards the SW part of the area." I am not sure if I can see this b-value increase. The figures that are mentioned show the 2018 epicenter, while Fig.13a that shows the 2016 event, this increase is not clear.

20) Rephrase the sentence "which in also retrieves the smoothing parameters for it."

---

## Author Comment (AC1)

1. **A more detailed description of the seismicity that is being studied and also, the fundamental parameters of the used catalogue, are missing. For example, maximum and minimum magnitude of the catalogue, threshold magnitude, generic errors in the hypocentre's locations (and therefore, time sensitivity of the located hypocentres), etc.**

**ANSWER TO REVIEWER:**

We agree with the reviewer that although some information was included in the manuscript regarding number of events, maximum and minimum magnitude of the catalogue, maximum and minimum depth, etc. a more detailed description is now included by adding the threshold magnitude and the generic errors of the location. The following modifications have been made:

Lines 211-219:

"In order to apply the proposed methodology, the Spanish earthquake catalogue (https://www.ign.es/web/ign/portal/sis-catalogo-terremotos) was filtered selecting the events in a 40 km radius circumference centered at Lorca's earthquake epicenter. Events have been selected from year 2000 to 2021 to have enough data to plot the b-value (Figure 7). This catalogue has a total of 2962 events with magnitudes between 0.8 Mw and 5.0 Mw (low to moderate earthquakes) and depths that range from 0 to 32.0 km (shallow seismicity). Before November 1997, epicentral location uncertainties were calculated with Hypo71 (Lee and Lahr, 1975) and specified as the so-called ERH (standard horizontal error, in km), however since November 1997, epicentral location uncertainties calculated by Evloc (Carreño-Herrero and Valero - Zornza, 2011) are reported as error ellipses at 90 % confidence level in the full-format catalogue. The epicentral location and the focal depth has uncertainties usually lower than 5 km within the Iberian Peninsula (González, 2017). The threshold magnitude for shallow seismicity is Mw 1.8."

Lines 227-234:

"It is noteworthy to highlight that all strong earthquakes registered have occurred at depths greater than 60 km, which have become the focus of research for this zone. The up-to-date catalogue of Romania can be found in the following address: http://www.infp.ro/index.php?i=romplus. Between 1990 and the end of 2013, locations were determined using the HYPOPLUS (Oncescu et al., 1996) program, a 1D velocity model and stations corrections. Starting with 2014, the earthquake location is obtained using Antelope software. In the present form, a single magnitude scale (Mw moment magnitude scale) is adopted for all the events. Different magnitude scales used before 2014 were converted into moment magnitude (Mw), based on calibration relations presented in Oncescu et al., 1999."

Lines 247-249:

The catalogue contains 6615 events with magnitude ranging from Mw 0.1 to Mw 6.0, 35 % have shallow depth, 19 % have intermediate depth and 46 % have deep depth. As we can see the most of them (65 %) are intermediate and deep seismicity. The cut-off magnitude for this catalogue is Mw 2.70.

References:

Oncescu M.C., Rizescu M., Bonjer K.P. (1996). SAPS – A completely automated and networked seismological acquisition and processing system, Computers & Geosciences 22, 89-97.

Oncescu M.C., Marza V.I., Rizescu M., Popa M. (1999). The Romanian earthquake catalogue between 984-1997, F.Wenzel et al. (eds.), Vrancea Earthquakes: Tectonics, Hazard and Risk Mitigation, 43-47, Kluwer Academic Publishers.

2. **Although throughout the paper the calculation of parameter b is frequently discussed, at no time is it indicated which method has been used for this purpose. The only reference is (line 91 to 93). From which it is assumed that the classic formula of Aki (1965) has been used. It should be clarified and if this has been the expression used, the authors should explain why the common improvement made from Utsu (1966) has not been used.**

**ANSWER TO REVIEWER:**

We agree with the comment and the manuscript has been improved to clarify that the maximum likelihood method using Utsu's formula (1966) has been used. Now lines 91 to 93 are written as:

> *"Another issue that has to be addressed is the method chosen for the estimation of the cut-off magnitude calculus. Recent work (Zhou et al., 2018) has shown that the characteristics of the seismic catalogue determine which algorithm suits better the cut-off or threshold magnitude calculus which is needed to calculate the b-value according to maximum likelihood method proposed by Aki (1965) and improved by Utsu (1966)."*

Reference:

Utsu, T.: A Statistical Significance Test of the Difference in b-value between Two Earthquake Groups, Journal of physics of the earth, 14, 37–40, 1966.

3. **Authors state that the exponential-type function fits to data better than the Gaussian (lines 183 and 184). It should be justified using the correlation coefficient from table 1, such as with table two they did.**

**ANSWER TO REVIEWER:**

We agree that there was a missing reference for the table 1 coefficient. The standard error of the model, S, has been deleted from the tables as the correlation coefficient, $R^2$, is more descriptive in terms of adjustment errors. The lines 183 and 184 have been changed as follows:

> *"The exponential-like function is a better fit for the inter-event distance distribution as it can be seen in both Figure 5 and Table 1, where the correlation coefficient - a measure of how much the points of the model function differ from those of the dataset -, $R^2$, is closer to 1 for the exponential-like function."*

4. **Abbreviation CPTI15 is not explained (line 106).**

**ANSWER TO REVIEWER:**

We agree that the definition of this acronym has not been presented in the text. The line 106 has been modified as follows:

> "...contained ==in half of the Parametric Catalogue of the Historical Italian earthquakes (CPTI15)== and obtained a smoothing parameter of 30 km for central Italy."

5. **Although the number and quality of the references appropriate and they are accessible by scientists, there exist other significant advances (that should be added) and it bring this manuscript in accord with the recent literature; for instance, (line 47 to 50) can be re-written as follow:**

> *"Recent studies have shown the importance of the so-called b-value regarding seismic risk assessment by relating its low values (depending on the tectonic regime and the area) to tectonic stress build-up (Gulia and Wiemer, 2010) Moreover, the conclusions of this work agree with tests conducted in laboratory scale (Wiemer and Schorlemmer, 2007).* **Therefore, the relationship demonstrated by De Santis et al. (2019) between b parameter and the Shannon Entropy has allowed the use of this thermodynamic variable as an indicator of the occurrence of an earthquake (Posadas et al., 2021, 2022); but, in addition, non-extensive entropy (Vallianatos et al. 2018, 2020) is also likely to be used in the same terms (Papadakis et al., 2015).** *Finally, Galiana-50 Merino et al. (2022) proved the viability of using radon measurements to estimate the daily seismic activity rate."*

*References:*

*De Santis, A., Abbattista, C., Alfonsi, L., Amoruso, L., Campuzano, S., Carbone, M., Cesaroni, C., Cianchini, G., De Franceschi, G., De Santis, A., Di Giovambattista, R., Marchetti, D., Martino, L., Perrone, L., Piscini, A., Rainone, M., Soldani, M., Spogli, L., Santoro, F., "Geosystemics View of Earthquakes", Entropy 21, 412-442 (2019).*

*Papadakis G, Vallianatos F, Sammonds P. A nonextensive statistical physics analysis of the 1995 Kobe, Japan earthquake. Pure ApplGeophys, 2015; 172:1923–31. https://doi.org/10.1007/s00024-014-0876-x.*

*Posadas, A., Morales, J., Ibáñez, J., Posadas-Garzon, A., Shaking earth: Non-linear seismic processes and the second law of thermodynamics: A case study from Canterbury (New Zealand) earthquakes, Chaos, Solitons and Fractals. Nonlinear Science, and Nonequilibrium and Complex Phenomena, 151, 2022. doi.org/10.1016/j.chaos.2021.111243.*

*Posadas, A., Morales, J., Posadas-Garzon, A., Earthquakes and entropy: Characterization of occurrence of earthquakes in southern Spain and Alboran Sea, Chaos: An Interdisciplinary Journal of Nonlinear Science, 31 (4), 2021. doi.org/10.1063/5.0031844.*

*Vallianatos, F., Michas, G., Papadakis., G., Nonextensive Statistical Seismology: An Overview. Complexity of Seismic Time Series. In Chelidze, T., Vallianatos, F., Telesca, L., editors. Complexity of Seismic Time Series: Measurement and Application. Elsevier, 2018, p25-59. https://doi.org/10.1016/B978-0-12-813138-1.00002-X.*

*Vallianatos, F., Michas, G., Complexity of Fracturing in Terms of Non-Extensive Statistical Physics: From Earthquake Faults to Arctic Sea Ice Fracturing, Entropy 2020, 22, 1194; doi:10.3390/e22111194.*

**ANSWER TO REVIEWER:**

The manuscript has been changed and the references have been added as they are fitting for the state-of-the-art introduction.

6. **Figure and table captions are too short. One should be able to fully understand the meaning of the figure or table without appealing to the body of the manuscript. For example, in table 1 caption it is not explained what R is and what S is (nor in the main text) or, in figures 2 and 3, definition of spatial cell-event distance and the inter-event distances should be indicated. On the whole, a more detailed, broad and comprehensive captions are needed.**

**ANSWER TO REVIEWER:**

We agree that explanation is due for parameters such as R, inter-event and spatial cell-event distances. The following changes have been made:

The $R^2$ parameter has been referenced in both the captions of the tables and in lines 183-185 as seen in answer to question 3. As for the inter-event and spatial cell-event distances, these quantities have been now defined in lines 122-125 as follows:

> *"First, it is necessary to study both the inter-event distance and the spatial cell-event distance distribution. The inter-event distance is the distance between any two events of the catalogue (in any of case studies the distances between all the event pairs will be calculated), as for the spatial-cell event distance, it is defined as the distance between a spatial grid cell and an event from the catalogue (as in the former definition the distances between all the spatial cells and all the events will be calculated)."*

The captions of figures 1-3 have been modified in order to be more descriptive:

> *"Figure 1. Frequency-magnitude plot for the Italian CPTI15 earthquake catalogue. A total of 56309 events have been used."*

> *"Figure 2. Histograms of the distances between every spatial cell and event pair (spatial cell-event distances) of the CPTI15 Italian earthquake catalogue at different time periods."*

> *"Figure 3. Histograms of the distances between every event pair (inter-event distances) of the CPTI15 Italian earthquake catalogue at different time periods."*

---

## Author Comment (AC2)

1. **Inter-event distance is usually a term describing the distance between successive earthquakes. In the paper, it seems that this term is used to describe the distance between each event and all the other events in the catalogue. This point needs some further clarification in the part where Equation 2 is given. Apparently, only the coordinates of the epicentres are used to estimate inter-event distances. Is there a reason why depths are not used in the calculations? Please explain.**

**ANSWER TO REVIEWER:**

We agree with the reviewer that the term inter-event distance is confusing so instead event-event distance will be used when referring to Euclidean distance between any pair of events of the catalogue. The definition of this parameter is now explained in lines 122-125 as follows:

> *"First, it is necessary to study both the* ==event-event== *distance and the spatial cell-event distance distribution. ==The event-event distance is the distance between any pair of events of the catalogue (in each of the case studies the distances between all the event pairs will be calculated), as for the spatial-cell event distance, it is defined as the distance between a spatial grid cell and an event from the catalogue (as in the former definition the distances between all the spatial cells and all the events will be calculated).=="*

In the case of Spain, the seismicity is shallow and the hypocentral and epicentral distances are similar. For the region of Vrancea in Romania, the catalogue was filtered so only the intermediate-deep earthquakes remain. This way, the mean depth is assumed to be homogeneous, resulting in this component having no effect in the calculations.

2. **How exactly the weight function (Equation 3) works in the calculations? The authors cite the paper of Taroni and Akinci (2021), but some further clarifications and examples are needed to inform the reader how the weight function affects the b-value estimations.**

**ANSWER TO REVIEWER:**

We agree with the comment and the manuscript has been modified to elaborate in this matter:

$$W = W_{SK} \cdot W_{SS} \tag{3}$$

155    Where $W_{SK}$ is the smoothing filter and $W_{SS}$ is the function used to add foreshocks and aftershocks into the b-value calculus. This weight function operates inside the expression of the b-value as defined by Utsu (1965) and adapted by Taroni et al. (2021b):

$$\hat{b} = \frac{1}{\left(\sum_{i=1}^{N} W_i \cdot (M_i - M_{min}) + \frac{\Delta}{2}\right) \cdot \log 10} \tag{4}$$

160    Where $N$ is the total number of events in the catalogue, $M$ is the magnitude of the event, $M_{min}$ is threshold magnitude and $\Delta$ is the binning of the magnitude in the catalogue. In these case studies the threshold magnitude does not change in a manner that can affect the b-value calculus, so no changes depending on time windows have been considered.

The uncertainty of this b-value has been calculated following the procedure of Taroni et al. (2021b) and it was derived by these authors following Aki (1965) work and applying the delta method (Dorfman, 1938) to take into account the weight function used in the b-value calculation:

165    $$\hat{\sigma}_{\hat{b}} = \hat{b} \cdot \sqrt{\sum_{i=1}^{N} W_i^2} \tag{5}$$

The next figure has been added so there's a visual comparison of the weight each event is given depending on the distance:

[Figure]

3.  **Provide some more information regarding the Italian seismic catalogue (source etc.)**

**ANSWER TO REVIEWER:**

The information about the Italian Seismic Catalogue has been complemented in the first paragraph in which is mentioned:

> "... by means of the maximization of the likelihood of the seismicity contained ==in half of the Parametric Catalogue of the Historical Italian earthquakes (CPTI15 - Release v1.5-July 2016- from Rovida et al. (2020))== and obtained a smoothing parameter of 30 km for central Italy."

And the information about the events in this catalogue can be found when the case study is presented:

> "The Italian catalogue comprises the events from 1960 to 2019 for all Italy. It amounts up to 56309 events, which can be described in terms of magnitude and depth. The depth of the events ranges from 0 to 30.0 km, so the seismicity considered for this area is shallow. As for the magnitudes, the minimum is 1.81 Mw, and the maximum is 6.81 Mw."

Reference:

Rovida, A., Locati, M., Camassi, R., Lolli, B., and Gasperini, P.: The Italian earthquake catalogue CPTI15, Bulletin of Earthquake Engineering, 18, 2953–2984, https://doi.org/https://doi.org/10.1007/s10518-020-00818-y, 2020.

4. **In Figure 3, the inter-event distance histograms are shown for different time periods. How were these time periods chosen by the authors? For instance, we see a 50-years interval followed by a 4-years and 1-year interval.**

**ANSWER TO REVIEWER:**

The time periods were chosen so the number of events in each histogram is equivalent (roughly the same order) so the trend can be analysed. The conclusions drawn about the distance distribution have been made over the stacked histogram which was the combination of all four histograms.

5. **The authors consider two functions, the Gaussian and exponential functions, to obtain the smoothing kernel. However, power-law functions (e.g., Abe and Suzuki, 2003 and Corral, 2006) have also been considered in the literature, with their significance on the spatial clustering of earthquakes. Did the authors check the adequacy of such functions? A histogram in log-log axes will assist to show if power-law regimes exist.**

**ANSWER TO REVIEWER:**

The functions chosen as smoothing kernels have been considered based on existing literature for the exponential-like function (Tormann et al., 2014) and mathematical significance, as the Gaussian kernel has direct relationship with the distance distribution by means of the $\pi$ and $\sigma$ parameters. The power-law functions are interesting in order to study the distance distribution and worth a broader study regarding the application of the proposed methodology. The scope of this work was to showcase the methodology and a couple of examples (case studies and functions) but perhaps in the future a work focused on a single case study with several functions could be written.

6. **In line 181, the authors say, "For this function to be fitted, the count distribution has been normalized". How exactly it is normalized?**

**ANSWER TO REVIEWER:**

In order to avoid using a parameter for the normalization of the exponential function, the counts of the distance distribution for each bin of the histogram have been divided by the sum of all the counts. That is how we normalized the counts for the distance distribution. We decided to point this out, as the second y axis in the figure depicts different scale for the values of the exponential-like function. Now the text reads:

> *"For this function to be fitted, the count distribution ==has been normalized by dividing the counts of each bin by the sum of counts in all the bins so the parameters can be used for the weight function calculus without the need of a normalization constant==."*

7. **With which method exactly are the b-values and their uncertainties calculated?**

**ANSWER TO REVIEWER:**

We agree with the comment and the manuscript has been improved to clarify that the maximum likelihood method using Utsu's formula (1966) has been used. Now lines 91 to 93 are written as:

> *"Another issue that has to be addressed is the method chosen for the estimation of the cut-off magnitude calculus. Recent work (Zhou et al., 2018) has shown that the characteristics of the seismic catalogue determine which algorithm suits better the cut-off or threshold magnitude calculus which is needed to calculate the b-value ==according to maximum likelihood method proposed by Aki (1965) and improved by Utsu (1966)==."*

The uncertainty of this b-value is taken from Taroni et al. (2021) and it was derived by these authors following Aki's (1965) work and applying the delta method (Dorfman, 1938) to take into account the weight function used in the b-value calculation. The text has been modified so this information is clearly indicated:

References:

Aki, K. Maximum likelihood estimate of b in the formula $\log N = a - bM$ and its confidence limits, Bull. Earthq. Res. Inst. 43, 237–239, 1965.

Dorfman, R. A note on the $\delta$-method for finding variance formulae, Biometrics Bull. 1, 129–137, 1938.

8. **Is the difference between the b-value spatial distribution for Italy obtained in this work (Figure 6) within the confidence limits with that of Taroni et al.? The authors provide only the percentage change.**

**ANSWER TO REVIEWER:**

The algorithm that generates the figure has been changed so only the spatial cells whose b-values are within the confidence limits defined by Taroni et al. uncertainty are plotted. Out of the 4074 points in the array of b-values only 10 are out of this 95 % confidence interval. The Figure 6 has been changed although no visible changes appear. The text has been modified so the explanation about 95 % the confidence interval is visible:

[Figure]

*"Only the spatial cell grids whose b-value is in the 95 % Confidence Interval (CI) with that of Taroni et al. (2021b) are plotted (only 10 spatial cells out of 4074 where outside of the 95 % CI). The difference between the two spatial maps is lower than 2 % in most of the country except for border areas in which the difference can rise up to a 15 % as it can be seen in Figure 7. This can be due to less data being available for the b-value calculus (border effect)."*

9. **For the Lorca case, are the coordinates in Figure 7 and 9 the same? In the first figure we see positive values and in the second negative values for Longitude. In addition, the authors say that they selected the events at 40 km radius circumference centered at Lorca's earthquake epicenter. However, in Figure 7 we only see a fraction in a much smaller area. The text and the figures should be consistent.**

**ANSWER TO REVIEWER:**

We agree that the format of the coordinates should be the same for all the figures in the text and that the map should showcase the area in which the b-value calculus is being made. For this reason, the map in the Figure 7 has been remade so there is consistency between all the representations of the area of interest.

**South-eastern Spain seismic catalogue (2000 - 2020)**

[Figure]

**10. Why do the authors show the 1579 – 2021 seismicity in Figure 7 after all? Only 2000 – 2021 seismicity is studied.**

**ANSWER TO REVIEWER:**

We agree that the representation of the whole catalogue is not relevant, so the figure has been changed in order to only show the part of the catalogue that is important to the case study. The figure is the one shown in the previous question.

**11. Why is year 2011 excluded from the analysis for the Lorca case?**

**ANSWER TO REVIEWER:**

Yes, we avoided using data from the year in which the earthquake occurred in order to have the most representative results on the tectonic stress setting. Nevertheless, in this case after a careful consideration we have decided to include all the data before the earthquake in 2011 (excluding the seismic series in which this main shock belongs) so more events are available to the b-value calculus. Also, the events after this seismic series have been used in the latter b-value maps. The periods are now 2000-2011 and 2011-2020. The figures have been changed accordingly:

[Figure]

**12. In the caption of Figure 11 in c), what is "down" stands for?**

**ANSWER TO REVIEWER:**

It should have been deleted as the figure was changed from a previous configuration. It has been deleted now.

**13. In Line 210, the 0 – 100 km distance range is mentioned, but the radius for the seismicity selection is 40 km. Please be precise.**

**ANSWER TO REVIEWER:**

We agree that this sentence should be corrected as the maximum possible distance is 80 km for a 40 km radius circle:

> *"As it can be seen, there is no clear distribution for the 0 – 80 km interval…"*

**14. In Line 214 correct date with data.**

**ANSWER TO REVIEWER:**

The line has been corrected:

*"…for this reason, the functions that have been fitted in d) use all the available ==data==."*

**15. The use of the words "in each spatial cell" in the caption of Figure 9 adds some confusion. Do the authors mean the total number of events used in the analysis?**

**ANSWER TO REVIEWER:**

The total number of events is also the number of events we use for calculating the b-value in each spatial. Instead of using a cut-off distance (which is equivalent to the event window in the b-value time series) as Tormann et al. (2014) did, we use the spatial kernel, i. e. the Gaussian or exponential-like function to weight down the influence of the events based on their distance to the spatial grid cell in which the b-value is being calculated. In short, the total number of events used in the analysis and the number of events used for the b-value calculus is the same, although the weight of most of the events in the b-value calculus of a certain spatial grid cell will be negligible. For the sake of avoiding confusion, now in the captions in which "*in each spatial cell*" now it will read "*==n events have been used in the b-value calculus==*" and how the weight function operates can be found in the methodology section following the comment 2.

**16. In Line 230, the authors say, "Taking into consideration that the region has suffered three earthquakes with Mw 5.5 in the last decade, then we have chosen the events from 2016 to 2020 in the Vrancea region." This sentence does not make much sense and the selection is not justified. Probably the sentence needs rephrasing.**

**ANSWER TO REVIEWER:**

Yes, we agree that this sentence indeed makes no sense, as the time periods considered range from 2000 to 2018. This is a mistake that has been corrected so it is coherent with subsequent figures.

*"Taking into consideration that the region has suffered ==two== earthquakes with Mw 5.5 in the last decade, ==we have chosen the events from 2000 to 2018== in the Vrancea region. The b-value mapping area is comprised inside the blue frame Figure 10, in which all the events of the catalogue ==for this period== are shown."*

**17. Add the b-values in Figure 11 or discuss them in the text. Why not similar plots are shown for the Lorca case, or the G-R fitting for the Italian catalogue?**

**ANSWER TO REVIEWER:**

The b-values for all the G-R law fits have been added. In the cases of the Italian and Spanish catalogue they have not been considered as both catalogues have been already filtered so only shallow seismicity is considered. In the case of the Vrancea region the catalogue contains all the depth spectra, and the most representative seismic activity is the one of intermediate and deep depth, that is why the G-R fitting was calculated

for each group of depths. This information has been added in the text as clarification in the Vrancea case as it should have been done:

*"The events can be plotted in the frequency-magnitude graph depending on their depth. In this case deep seismicity accounts for intermediate and deep events (> 50 \unit{km} depth) and shallow seismicity for those with depth lesser than 50 \unit{km}. In this case, the catalogue has not been filtered prior calculus (the Italian and Spanish only contain shallow seismicity; hence they do not require a G-R fit to discriminate the different seismic settings)."*

*"Figure 11 represents the magnitude-frequency distribution of the earthquakes at different depths and the tendency of the G-R law. ==A b-value of 2.00 for the shallow seismicity has been found and a b-value of 0.91 has been calculated for the deep seismicity==. The different slopes indicate that separate catalogues should have been used to compute the spatial b-value distribution for each depth range. As the moderate to large earthquakes have a depth greater than 50 km, then we will consider only intermediate and deep seismicity for the study."*

**18. Again, the authors exclude year 2016 for Vrancea. Is this a form of "manual declustering" excluding major aftershock sequences?**

**ANSWER TO REVIEWER:**

Yes, we avoided using data from the year in which the earthquake occurred in order to have the most representative results on the tectonic stress setting. Nevertheless, in this case after a careful consideration we have decided to include all the data before the earthquake in 2016 (excluding the seismic series in which this main shock belongs) so more events are available to the b-value calculus. Also, the events after this seismic series have been used in the latter b-value maps. The periods are now 2000-2016 and 2016-2018. The results are now updated and shown in the remark number 20.

**19. Rephrase the sentence "Both b-value maps (with the different kernels) depict an increase of the b-value in zone where the earthquake of 2016 (Figure 13c and Figure 13d)**

**ANSWER TO REVIEWER:**

We changed the phrasing as it was confusing and tried to make a comparison using Figure 13 and Figure 14 so the b-value changes can be identified in a more direct way.

*"==The b-value maps in Figure 13c and Figure 14c when compared with those in Figure 13a and Figure 14a allow to identity an increase in the b-value near the epicentre of the earthquake of 2016== which could indicate tectonic stress relief and a slight b-value decrease towards the SW part of the area."*

**20. Which could indicate tectonic stress relief and a slight b-value decrease towards the SW part of the area." I am not sure if I can see this b-value**

**increase. The figures that are mentioned show the 2018 epicenter, while Fig.13a that shows the 2016 event, this increase is not clear.**

**ANSWER TO REVIEWER:**

We agree that the mark for all the epicenters should be present in all the subfigures, so the comparison is clear. The figure 13 has been modified so the shift in the tectonic stress build-up area and the slight increase in the b-value around the 2016 epicenter can be seen:

[Figure]

The figure 14 has been changed as well so the symbology is coherent:

[Figure]

**21. Rephrase the sentence "which in also retrieves the smoothing parameters for it."**

**ANSWER TO REVIEWER:**

We agree that the phrasing is confusing, so the manuscript has been changed as follows:

> "*Moreover, it avoids the arbitrary selection of a smoothing kernel by fitting a function to the distance distribution and obtaining the parameters for it.*"